# Direct thermal charging cell for converting low-grade heat to electricity

Xun Wang [1], Yu-Ting Huang[1], Chang Liu[1], Kaiyu Mu[1], Ka Ho Li[1], Sijia Wang [1], Yuan Yang [2], Lei Wang[1], Chia-Hung Su[3] & Shien-Ping Feng [1,4]

Efficient low-grade heat recovery can help to reduce greenhouse gas emission as over 70% of primary energy input is wasted as heat, but current technologies to fulfill the heat-to-electricity conversion are still far from optimum. Here we report a direct thermal charging cell, using asymmetric electrodes of a graphene oxide/platinum nanoparticles cathode and a polyaniline anode in $Fe^{2+}/Fe^{3+}$ redox electrolyte via isothermal heating operation. When heated, the cell generates voltage via a temperature-induced pseudocapacitive effect of graphene oxide and a thermogalvanic effect of $Fe^{2+}/Fe^{3+}$, and then discharges continuously by oxidizing polyaniline and reducing $Fe^{3+}$ under isothermal heating till $Fe^{3+}$ depletion. The cell can be self-regenerated when cooled down. Direct thermal charging cells attain a temperature coefficient of 5.0 mV K$^{-1}$ and heat-to-electricity conversion efficiency of 2.8% at 70 °C (21.4% of Carnot efficiency) and 3.52% at 90 °C (19.7% of Carnot efficiency), out-performing other thermoelectrochemical and thermoelectric systems.

[1] Department of Mechanical Engineering, The University of Hong Kong, Pokfulam Road, Hong Kong, China. [2] Department of Applied Physics and Applied Mathematics, Columbia University, New York, NY 10027, USA. [3] Department of Chemical Engineering, Ming Chi University of Technology, New Taipei City 24301, Taiwan. [4] The University of Hong Kong-Zhejiang Institute of Research and Innovation (HKU-ZIRI), Hangzhou 311300 Zhejiang, China. Correspondence and requests for materials should be addressed to S.-P.F. (email: hpfeng@hku.hk)

Energy conversion from primary energy carriers to the final energy of use is subject to considerable losses equivalent to ~72% of the global primary energy consumption[1]. The major loss is identified as waste heat, and specifically 63% of which occurs as low-grade heat below 100 °C[2,3]. Efficient recovery of low-grade heat is vital to abate greenhouse-gas emissions and also promises potential economic and environmental benefits. Current technologies such as solid-state thermoelectric devices (TEs) and liquid-based thermoelectrochemical cells (TECs) are capable of converting heat into electricity, but their conversion efficiency ($\eta_E$) is either too low or system is too complex for economical deployment. With current materials, solid-state TEs working within 100 °C have an $\eta_E$ less than 2%[4].

Liquid-based TECs are promising for low-grade heat harvesting as their temperature coefficient ($\alpha = \frac{\partial V}{\partial T}$, $V$ is the electrode voltage, $T$ is the temperature) is one order of magnitude higher than those of TEs (Seebeck coefficient of 100–200 μV K$^{-1}$)[5,6]. The electrochemical system offers the opportunity to engineer entropy changes, thermal and electrical transport. Conventional thermogalvanic cell (TGC) is operated under a temperature gradient, where an aqueous $Fe(CN)_6^{3-}/Fe(CN)_6^{4-}$ redox couple sandwiched between two Pt or carbon-based electrodes serves as entropy carrier and produces the $\alpha$ of −1.4 mV K$^{-1}$, yielding an $\eta_E$ of 0.1–0.5% (0.4–4% of Carnot efficiency $\eta_{Carnot}$)[6–9]. TGCs suffer the poor ionic conductivity of liquid electrolyte; although shortening the distance between two electrodes could improve the conductivity, it inevitably compromises $\eta_E$ because the additional energy is required to maintain the temperature difference. Alternatively, thermally regenerative electrochemical cycle (TREC) can be operated between hot and cold reservoirs that alternates in a thermal cycle, where cells are charged and discharged at different temperatures. TREC is configured as a pouch cell with short distance between two electrodes to greatly improve the electrolyte conductance, which made of a copper hexacyanoferrate (CuHCF) cathode and a Cu/Cu$^{2+}$ anode exhibits an $\alpha$ of −1.2 mV K$^{-1}$ and a high $\eta_E$ of 3.7% (21% of $\eta_{Carnot}$) when operating between 60 °C and 10 °C[10]. Nevertheless, the reliance on external electricity for the charging process complicates the system and limits its practical adoption. A charging-free TREC can be realized by carefully pairing the electrodes with the opposite voltages at low and high temperatures. Thus, the two electrochemical processes at both low and high temperatures in a cycle can be discharged by current flow in reverse direction, but the efficiency lower than 1% (10% of $\eta_{Carnot}$) and the use of expensive ion-selective membrane are concerns[11]. Similarly, thermally regenerative ammonia battery (TRAB) undergoes a thermal cycle between 56 °C and 23 °C, producing an $\eta_E$ of 0.53% (13% of $\eta_{Carnot}$)[12,13]. Yet, the introduction of ammonia stream is a serious concern regarding the leakage, stability, and safety. Recently, continuous electrochemical heat engine has also been conducted in the form of redox flow battery (RFB) with flowing electrolytes in redox reactions at different temperatures, which achieves an $\eta_E$ of 1.8% (15% of $\eta_{Carnot}$)[14].

In this work, we report the direct thermal charging cell (DTCC), which can be directly charged by heating without the need of external electricity and then electrically discharged at a high temperature ($T_H$); the system can be self-regenerated when cooled down at room temperature ($T_L$). Here, the energy conversion cycle with isothermal charging/discharging operation at $T_H$ and regenerative mode at $T_L$ is fundamentally different from the existing TEC technologies; the detailed comparison of features for the reported TECs and our DTCC is shown in Supplementary Table 1. The DTCC, using asymmetric electrodes inclusive of a graphene oxide/platinum nanoparticles (GO/PtNPs) cathode and a polyaniline (PANI) anode incorporating with an aqueous Fe$^{2+}$/Fe$^{3+}$-based redox electrolyte, is thermally charged via the temperature-induced pseudocapacitive effect of GO and the cooperative thermogalvanic effect of Fe$^{2+}$/Fe$^{3+}$, and then discharged successively under the occurrence of redox reactions of PANI and Fe$^{2+}$/Fe$^{3+}$. Our DTCC exhibits a markedly high $\alpha$ of 5.0 mV K$^{-1}$ and attains $\eta_E$ of 2.8% at 70 °C (21.4% of $\eta_{Carnot}$) and 3.52% at 90 °C (19.7% of $\eta_{Carnot}$) by establishing a thermal-induced potential gradient across asymmetric electrodes and then cycling via chemical regeneration of electrode and electrolyte materials.

## Results

**Working principle of DTCC.** The DTCC is configured as a pouch cell with asymmetric electrodes, a capacitor-type cathode of GO/PtNPs, and a battery-type anode of PANI, in aqueous FeCl$_2$/FeCl$_3$ redox electrolyte (Fig. 1a, Supplementary Fig. 1). The pouch cell configuration shortens the distance between two electrodes and greatly improves the ionic conductivity. Initially (Fig. 1b), a built-in open circuit voltage ($V_{OC}$) is observed due to the differential of electrochemical potentials ($\Delta V_0$) between GO/PtNPs and PANI electrodes at $T_L$. Then, DTCC undergoes three stages corresponding to thermal charging, electrical discharging, and self-regeneration in one cycle. When DTCC is heated from $T_L$ to $T_H$ in open circuit condition (stage 1), a thermal-induced voltage is built up by two parts inclusive of the temperature-induced pseudocapacitive effect of GO and the thermogalvanic effect of Fe$^{2+}$/Fe$^{3+}$ while PANI has little contribution to the increase voltage. First, heating enhances the chemisorption of protons on the oxygen functional groups of GO and causes the pseudocapacitive reaction occurring at the GO–aqueous interface (Fig. 1c); this rapid surface Faradaic reaction results in the electrochemical potential change of GO, which generates an increased $V_{OC}$[15–17]. Here, the small positive $\alpha$ of PANI causes a slight potential shift of PANI anode. Second, the reduction reaction of Fe$^{3+}$ happens under the catalysis of GO/PtNPs at $T_H$ due to the positive $\alpha$ of Fe$^{2+}$/Fe$^{3+}$, which further enhances the $V_{OC}$ (Fig. 1d)[18,19]. When an external load is connected (stage 2), the DTCC is discharged continuously at $T_H$, where the discharge capacity is mainly attributed to the oxidation of PANI anode and the simultaneous reduction of Fe$^{3+}$ on catalytic GO/PtNPs cathode (Fig. 1e). The cell $V_{OC}$ drives the oxidation of PANI anode to produce electrons through the external circuit; the majority of electrons are then carried by the reduction reaction of Fe$^{3+}$ to Fe$^{2+}$ in cathode side, which protects the functional groups of GO from being reduced and significantly enhances the discharge capacity of DTCCs. The discharging process ceases until the cell voltage decreases to $\Delta V_0$, as the Fe$^{3+}$ is depleted to cause the electron accumulation in cathode. At stage 3 (Fig. 1f), the oxidized PANI would react with Fe$^{2+}$ in an acidic electrolyte at $T_L$ to chemically regenerate PANI and Fe$^{3+}$; this synergistic self-regeneration mechanism allows the cyclability of DTCCs. Temperature–entropy ($T$–$S$) diagram and the corresponding chemical cycle are illustrated in Fig. 1g, h, respectively. During thermal charging, the entropy increases with the isobaric heating from $T_L$ to $T_H$. Besides, the chemisorption of protons increases the number of molecules at the GO–aqueous interface, leading to a spatial entropy increase at the interfacial region. During electrical discharging at $T_H$, the entropy increases along with oxidizing PANI and reducing Fe$^{3+}$. On one hand, the oxidation of PANI conduces to an entropy increase due to the release of proton; on the other hand, the less charged Fe$^{2+}$ core leads to a more disorder solvation shell as compared to Fe$^{3+}$, resulting in entropy increase when Fe$^{3+}$ is reduced to Fe$^{2+}$ (ref. [20]). Lastly, the entropy decreases and the chemical cycle is completed at $T_L$ with the reversible thermo-pseudocapacitive effect of GO and the self-regeneration of PANI and Fe$^{3+}$ (Supplementary Note 1).

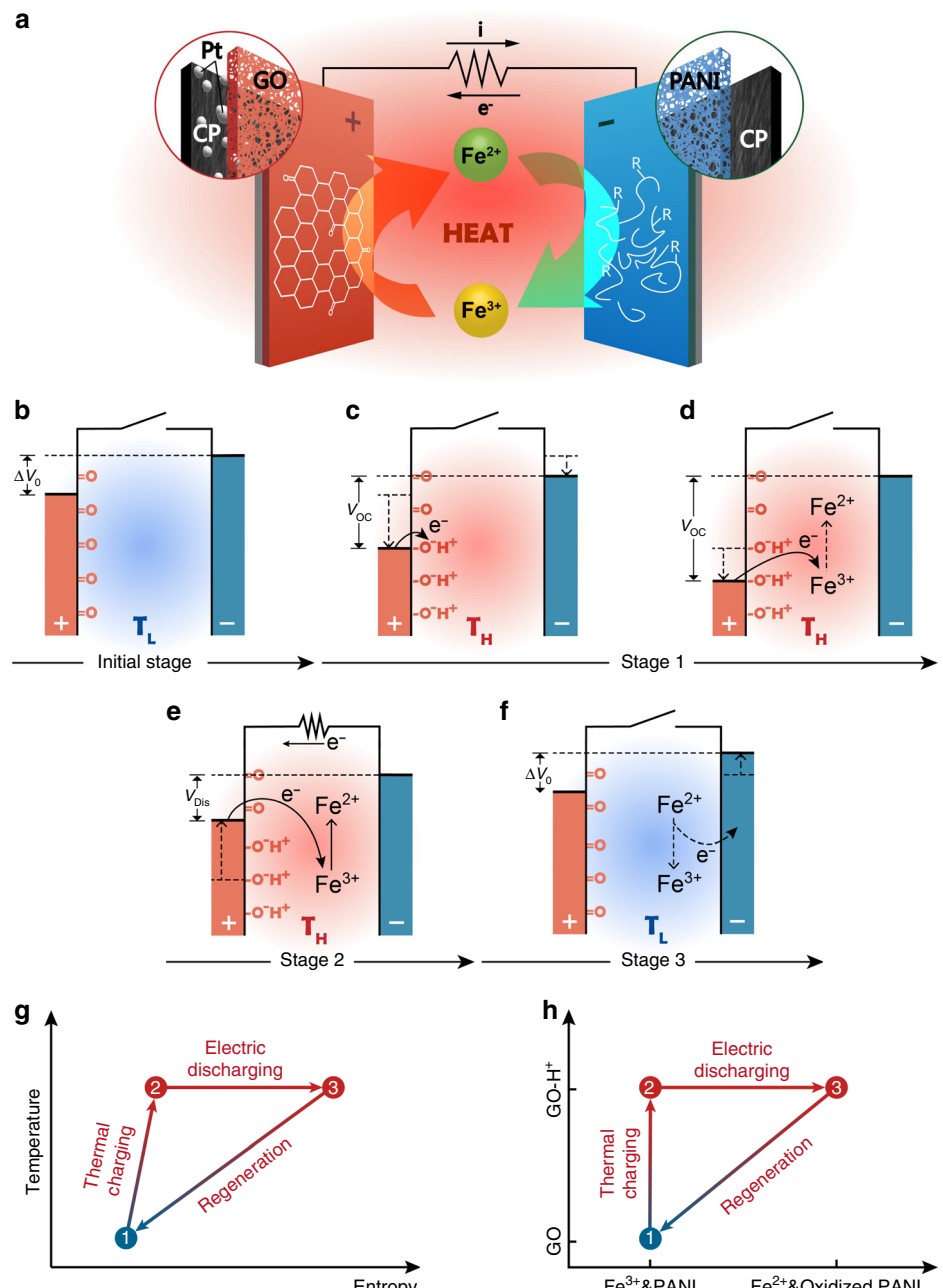

**Fig. 1** Working principle. **a** Scheme of DTCC consisting of GO/PtNPs cathode and PANI anode (with carbon paper (CP) substrate) in $Fe^{2+}/Fe^{3+}$ redox electrolyte. **b** Built-in voltage ($\Delta V_O$) based on the differential of electrochemical potentials between two electrodes at $T_L$. Heat-to-electricity conversion undergoes three stages. Stage 1: thermal charging process, DTCC is heated up from $T_L$ to $T_H$ in open circuit condition; **c** pseudocapacitive reactions occur at the GO–aqueous interface, generating a thermal-induced voltage, and **d** the $V_{OC}$ is further enlarged with the simultaneous reduction reaction of $Fe^{3+}$ at $T_H$. Stage 2: **e** electrical discharging process, DTCC produces current via the oxidation of PANI and the reduction of $Fe^{3+}$ at $T_H$. Stage 3: **f** self-regeneration process, PANI and $Fe^{3+}$ are chemically regenerated at $T_L$. **g** Temperature–entropy ($T$–$S$) diagram and **h** corresponding chemical cycle of DTCC

**Temperature-induced pseudocapacitive effect.** Conway and Gu investigated the kinetics of $Cl^-$ adsorption on polycrystalline Pt, and found that the voltage slowly rose to a plateau upon exposure of Pt to KCl solution without applying an electrical potential[21]. Qiao et al. observed a transient temperature-dependent voltage across two identical carbon electrodes when they were placed in high and low temperatures, respectively[22,23]. We discovered a voltage generated in which a commercial supercapacitor could be thermally charged after an electrical charge/discharge step[17]. It is believed that the increased voltage was induced by surface redox reaction at solid–liquid interface, related to ion adsorption/

desorption and electrode functionalities. Here, GO was chosen as active electrode as it contains abundant oxygen functional groups (e.g., carbonyl ($C=O$) and carboxyl ($O-C=O$))[24,25], which facilitate the chemisorption of protons in the electrolyte to generate pseudocapacitance[17,26]. Figure 2a exhibits that the $V_{OC}$ is a function of the operating temperature and four cells with different anodes and electrolytes were examined and compared each other. Three cells, consisting of GO or GO/PtNPs cathode with potassium chloride (KCl) electrolyte and one has a Ti anode (GO|KCl|Ti) while the other two have PANI anodes (GO|KCl|PANI and GO/PtNPs|KCl|PANI), have the similar $\alpha$ of 3.5, 3.3, and

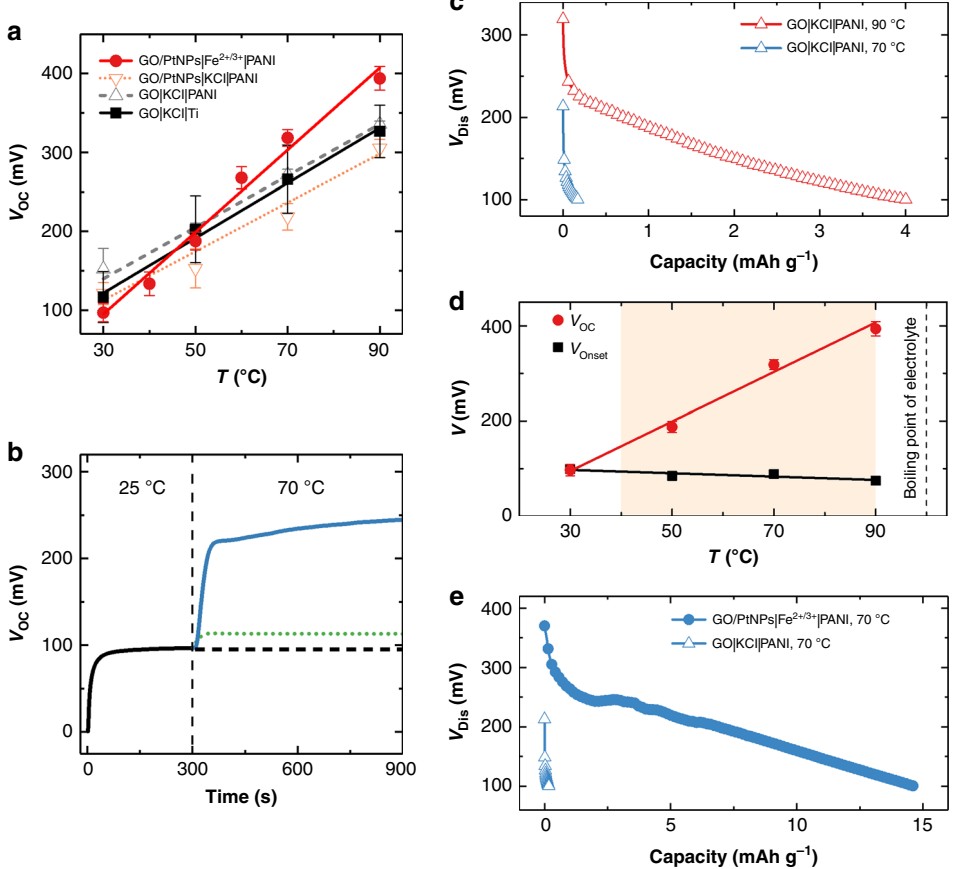

**Fig. 2** Thermal-induced voltage and electricity generation. **a** $V_{OC}$ versus temperature for four cells, GO|KCl|Ti, GO|KCl|PANI, GO/PtNPs|KCl|PANI, and GO/PtNPs|Fe$^{2+}$/Fe$^{3+}$|PANI. **b** $V_{OC}$ of GO|KCl|Ti cell at 25 °C and then heated to 70 °C, the black dashed line indicates $V_{OC}$ at 25 °C and green dotted line indicates the change of the $V_{OC}$ only arising from the EDL expansion at 70 °C. **c** Discharging curve of GO|KCl|PANI cell at 70 °C and 90 °C. **d** $V_{OC}$ of DTCC and $V_{Onset}$ of PANI versus temperature, orange region indicates the operating temperature range of DTCCs. **e** Discharging curves of GO|KCl|PANI cell and GO/PtNPs|Fe$^{2+}$/Fe$^{3+}$|PANI DTCC at 70 °C. Error bars were obtained from the standard errors of mean of independent samples

3.1 mV K$^{-1}$, respectively, indicating the decisive role of GO and ruling out the contribution of PANI to voltage generation (Supplementary Note 2, Supplementary Fig. 2a–c). A built-in $V_{OC}$ of 97 mV in GO|KCl|Ti was observed at 25 °C, and increased significantly to 244 mV after heating at 70 °C for 10 min (Fig. 2b); the green dotted line illustrates the voltage rise (31 mV) of electrical double layer (EDL) expansion based on calculation[27] (Supplementary Note 3), which only contributes a small part of the overall thermal-induced voltage. Hence, the rest voltage increase (131 mV), therefore, originates from other surface reactions at the GO–aqueous interface. The discharging curve unveils the dependency of the discharging voltage ($V_{Dis}$) and discharge capacity on $T_H$; the capacitive behavior of GO is observed that $V_{Dis}$ decreases linearly along with discharging process (Fig. 2c). Electrochemical impedance spectroscopy (EIS) was then conducted using a symmetric cell (GO|KCl|GO) to investigate the capacitive behavior at the GO–aqueous interface toward the temperature change. The total capacitance of GO–aqueous system increases proportionally with rising temperatures and shows a more pronounced resistance behavior when increasing temperatures (Supplementary Note 4, Supplementary Fig. 3), indicating the emergence of Faradaic reactions at $T_H$. The findings described above evidence the temperature-induced pseudocapacitive effect of GO. The temperature profile and $V_{OC}$ of the GO|KCl|Ti cell during cyclic heating/cooling processes shows that the voltage was generated with increasing temperatures and decreased with decreasing temperatures, indicating that the

proton adsorption/desorption on oxygen functional groups at the GO–aqueous interface is reversible and reproducible in this operating temperature range[28] (Supplementary Fig. 4).

X-ray photoelectron spectroscopy (XPS) characterization shows that the as-prepared GO is equipped with oxygen-containing groups inclusive of aromatic rings (C−C), epoxides and hydroxyl (C−O), C = O, and O−C = O groups (Supplementary Fig. 5a). After repetitive charging/discharging processes, the signals of C = O and O−C = O groups were dramatically reduced (Supplementary Fig. 6b), correlating to the undetectable thermal-induced voltage. On the other hand, an enhanced thermal-induced voltage was obtained if the GO electrode was treated with oxygen plasma to produce more C = O groups (Supplementary Figs. 5c and 6). Conclusively, the C = O and O−C = O functional groups in the GO play a key role in the thermal-charging mechanism where the temperature rise induces a fast and reversible Faradaic process arising from the chemisorption of proton on oxygen functional groups of GO[15,29–31] (Supplementary Note 5, Supplementary Fig. 7).

**PANI anode and Fe$^{2+}$/Fe$^{3+}$ redox couple**. In DTCC, the utilization of conductive polymer PANI and Fe$^{2+}$/Fe$^{3+}$ redox couple increases the overall reaction entropy change and guarantees a sustainable current. The leucoemeraldine form of PANI with doping of H$^+$ or Cl$^-$ was used, which can switch between different redox forms[32,33]. The anodic onset potential ($V_{Onset}$) of

PANI decreases with increasing temperatures (Fig. 2d) so that the current can be produced when $V_{OC} > V_{Onset}$ and PANI is oxidized from protonated leucoemeraldine to emeraldine salt[32]. The operating temperature of DTCCs is in the range of 40–90 °C, as the boiling point of the aqueous electrolyte determines the upper-temperature limit. The operating temperature window could be further extended by employing organic or ionic electrolytes[34]. Here, $Fe^{2+}/Fe^{3+}$ is chosen because of its intrinsic positive $\alpha$ so that heating facilitates the reduction reaction instead of the oxidation reaction of other commonly used redox couples (e.g., $Fe(CN)_6^{4-}/ Fe(CN)_6^{3-})$[7]; the electrons transfer from the GO/PtNPs to the lowest unoccupied molecular orbital (LUMO) of $Fe^{3+}$ during heating conduces to an enlarged $V_{OC}$. Thus, GO/PtNPs|$Fe^{2+}/Fe^{3+}$|PANI DTCC reaches a markedly high $\alpha$ of 5.0 mV $K^{-1}$ in thermal charging stage (Fig. 2a, Supplementary Fig. 2c). In the discharging process, with the aid of catalytic PtNPs, the $Fe^{3+}$ effectively carries the electrons from GO/PtNPs cathode and is reduced to $Fe^{2+}$, which prevents the oxygen functional groups of GO from being reduced and consumed to retain the thermal-induced voltage and thus sustains the oxidation of PANI (Supplementary Note 5, Supplementary Fig. 8). The catalytic effect of GO/PtNPs is described in Supplementary Note 6 (Supplementary Figs. 9 and 10). Notably, to compare with the control cell without using $Fe^{2+}/Fe^{3+}$, the reduction of $Fe^{3+}$ to $Fe^{2+}$ dominates the greater discharging capacity of DTCCs (Fig. 2e). The discharging process ceases when the $V_{OC}$ becomes smaller than the initial $\Delta V_0$ (~100 mV) because of the depletion of $Fe^{3+}$ in the electrolyte and the electron accumulation in the cathode. When cooled down, the $Fe^{2+}$ tends to be oxidized to $Fe^{3+}$ while the emeraldine salt of PANI is cathodic electroactive in the acidic electrolyte[35]. Therefore, the oxidized PANI can be reduced in the presence of a high concentration of $Fe^{2+}$ and $H^+$ at $T_L$, which synergistically regenerates PANI and $Fe^{3+}$ and allows the cyclability of DTCCs (Supplementary Note 7, Supplementary Fig. 11)[36].

**Heat-to-electricity conversion of DTCCs.** DTCC is configured as a three-electrode pouch cell with a titanium (Ti) reference electrode to measure the potential of each electrode independently (Supplementary Fig. 12). The temperature profile and voltage of DTCC during thermal charging and electrical discharging under the current density of 4.8 mA $g^{-1}$ at $T_H = 70$ °C are shown in Fig. 3a, b, respectively. The potential profile of the GO/PtNPs cathode ($V_{GO/PtNPs}$) is in good accordance with the full cell voltage (Fig. 3c). In contrast, the potential of PANI anode ($V_{PANI}$) has little voltage change (Fig. 3d), consistent with the observation in Fig. 2a. Figure 3e shows the current density–voltage ($I$–$V$) and the volumetric power density–voltage ($P$–$V$) curves of the DTCC operating at different temperatures. A higher $T_H$ offers a larger current density at a specific output voltage. The maximal volumetric power densities are 289, 856, and 3345 W $m^{-3}$ corresponding to 50 °C, 70 °C, and 90 °C, respectively, which are 1.4, 2.4, and 3.5 times higher than those of GO|KCl|PANI cell (Supplementary Fig. 13). Figure 3f demonstrates the cyclability of the DTCC, in which one cycle involves the thermal charging and electrical discharging processes at $T_H = 70$ °C and the self-regeneration process at $T_L = 25$ °C. At the first cycle, the $V_{OC}$, the average volumetric and gravimetric power density are 271 mV, 1344 W $m^{-3}$, and 12 W $kg^{-1}$, respectively, which decrease by less than 20% after 20 cycles, demonstrating decent reversibility of the temperature-dependent characteristics and good reproducibility of the device performance. Here, it is noted that the oxygen functional groups of GO are still consumed and PANI is gradually oxidized at each cycle until their depletion, which would limit the long-term cycle number (Supplementary

Note 8, Supplementary Figs. 14 and 15). Therefore, further research with the optimization of electrode materials, electrolyte, and cell packaging is needed to improve the cyclability of DTCCs. Additionally, DTCCs are applicable to an intermittent heat source, where the $V_{OC}$ and $V_{Dis}$ show the good reproducibility toward irregular heating (Supplementary Note 9, Supplementary Fig. 16); the rapid thermal response of DTCCs allows an immediate generation of voltage and electricity.

**Device performance and demonstration.** To evaluate the heat-to-electricity conversion efficiency of the DTCC, the $\eta_E$ is estimated as the output electrical work ($W$) divided by input thermal energy inclusive of the $Q_H$ for heating process and $Q_{iso}$ for heat absorbed at $T_H$, which can be expressed as[10,11]

$$\eta_E = \frac{W}{Q_H + Q_{iso}} = \frac{\int V dq}{(1 - \eta_{HX})\sum mC_P\Delta T + T_H\Delta S}, \quad (1)$$

where $q$ is the discharge capacity, $\eta_{HX}$ is the efficiency of heat recuperation, $m$ is the mass of the active materials of electrodes and electrolyte (e.g., GO, PANI, electrolyte), $C_P$ is the specific heat, $\Delta T$ is the temperature difference between the operating temperature and room temperature (e.g., $\Delta T = 90–25 = 65$ °C), and $\Delta S$ is the reaction entropy change, which can be retrieved from $\alpha$. Details of calculation are shown in Supplementary Note 10 and Supplementary Fig. 17.

The $\eta_E$ of DTCC reaches 2.8% at 70 °C and 3.52% at 90 °C (Fig. 4a), which is among the highest efficiency of the existing direct energy conversion technologies in this temperature range[6,11–14,37–39]. Notably, the $\eta_E$ of the DTCC is catching up with that of the solid-state TE with a figure of merit value of 1 (ref. [40]). Besides, the $\eta_E$ could be further improved to 4.5% and above if 50% of heat recuperation is employed[10]. The ratio of $\eta_E$ to $\eta_{Carnot}$ ($\eta_E/\eta_{Carnot}$) versus volumetric power density (W $m^{-3}$) is presented in Fig. 4b. The DTCC obtains 21.4% of $\eta_{Carnot}$ at 70 °C ($\eta_{Carnot} = 13.1\%$) and 19.7% of $\eta_{Carnot}$ at 90 °C ($\eta_{Carnot} = 17.9\%$). DTCC exhibits a superior volumetric power density over TGCs since the isothermal heating operation greatly shortens the distance between two electrodes; contrarily in TGCs, the reliance on the temperature gradient across hot and cold electrodes leads to a large volume and thus decrease the volumetric power density. The DTCCs also demonstrate a higher $\eta_E/\eta_{Carnot}$ ratio versus gravimetric power density (W $kg^{-1}$) compared to that of TREC, and their gravimetric power densities are comparable to those of fuel cells (Supplementary Note 11, Supplementary Fig. 18)[41].

Besides the excellent heat-to-electricity efficiency and power density, DTCCs have uniqueness and advantages for practical application including a wide operation window, isothermal and continuous charging/discharging process, low-cost and simple system, and the ability to form stacks of cells. For instance, one DTCC was able to charge a commercial supercapacitor to 0.15 V at 70 °C (Supplementary Note 12, Supplementary Fig. 19). In Fig. 4c, six DTCCs were stacked in series and heated up in hot water with a temperature ~70 °C to immediately trigger the phase transition (from transparent to dark) of an electrochromic smart window (2 V, 1 μA). In Fig. 4d, the bendable DTCCs can be placed along the surface of running compressor with a temperature of about 55 °C to recover the waste heat and power an organic light-emitting diode display (OLED, 3.5 V, 0.5 mA), where a HKU(The University of Hong Kong) logo was lighted up. All demonstrations can be seen in Supplementary Movie 1. The excellent performance and easy adoption for various conditions suggest that DTCCs may find many applications beyond the aforementioned scenarios.

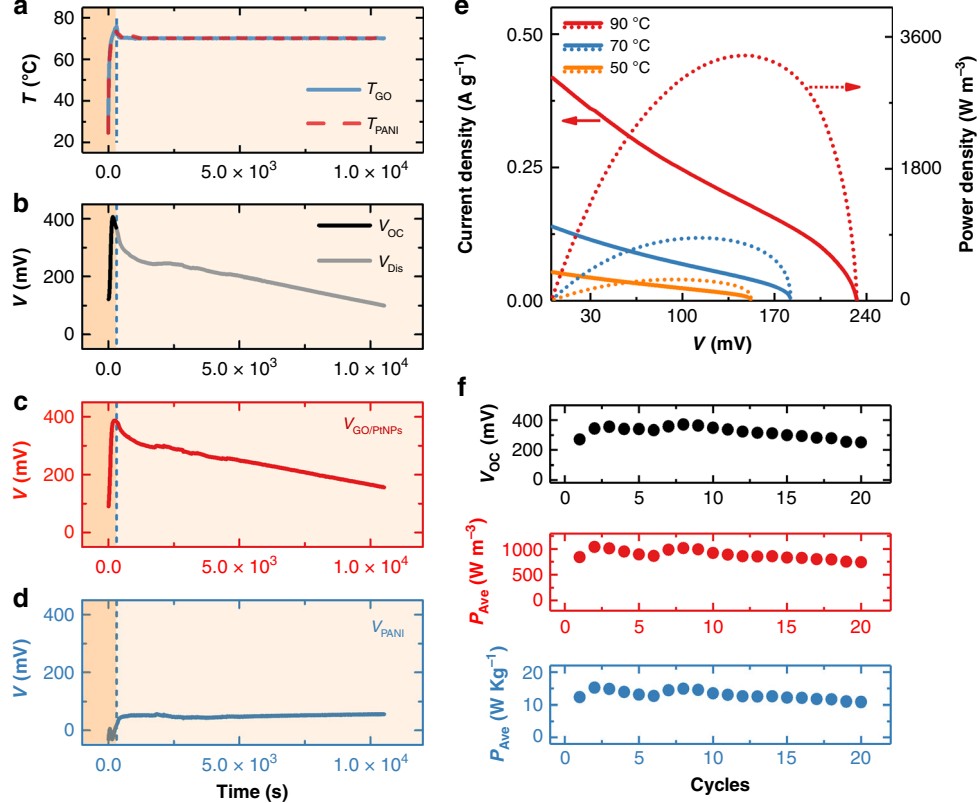

**Fig. 3** Heat-to-electricity conversion. **a** Temperature profiles at two electrodes (isothermal heating operation), **b** DTCC full cell voltage, **c** potential of GO/PtNPs cathode, and **d** potential of PANI anode during thermal charging (orange region) in open circuit condition and continuous electrical discharging (light orange region) at 70 °C. **e** Current density (solid line) and volumetric power density (dotted line) of DTCC with respect to voltage at various temperatures. **f** $V_{OC}$ (black), average volumetric power density (red), and average gravimetric power density (blue) of DTCC versus cycle numbers

## Discussion

In this work, we report efficient conversion of low-grade heat to electricity with an electrochemical cell, which consists of asymmetric GO/PtNPs and PANI electrodes with $Fe^{2+}/Fe^{3+}$ redox electrolyte. The DTCC is unique due to its dependence on a thermal-induced potential gradient and chemical cycle, which allows the isothermal heating operation during the entire charging and discharging processes instead of using temperature gradient in geometric configuration or thermal cycle. The DTCC achieves a high $\alpha$ of 5.0 mV K$^{-1}$ and a high $\eta_E$ of 2.8% at 70 °C (equivalent to 21.4% of $\eta_{Carnot}$) and 3.52% at 90 °C (equivalent to 19.7% of $\eta_{Carnot}$), which is at the forefront performance compared with the existing TECs and TEs in the low-grade heat regime. The systematic investigation of the thermal response in DTCCs has been conducted on the cathode, anode, and the electrolyte, of which the temperature-induced pseudocapacitive reactions on $C=O$ and $O-C=O$ functionalities of GO and the thermogalvanic effect of $Fe^{2+}/Fe^{3+}$ are responsible for the generation of thermal-induced voltage, and the PANI anode and $Fe^{2+}/Fe^{3+}$ redox couple are of importance to produce current by switching between their redox forms. When cooled down, the reversible thermo-pseudocapacitive effect of GO and the synergistic self-regeneration of PANI and $Fe^{3+}$ completes the chemical cycle of DTCCs. The great applicability and fast thermal response of DTCCs have been demonstrated in some practical scenarios. Notably, the DTCCs immersed in a hot water and placed on a running compressor can immediately power an electrochromic smart window and light up an OLED. This is the first demonstration of heat-to-electricity conversion undergoing isothermal heating and chemical regeneration, which revolutionizes the design of thermoelectrochemical systems; it is fundamentally different from the state-of-the-art technologies with power generation coupled to temperature differential. Further studies with robust high-$\alpha$ material and organic/gel-type redox electrolyte in DTCCs will better improve the cyclability and widen the operating temperature to utilize medium-grade heat or body heat-powered technologies[42].

## Methods

**Material synthesis and electrode preparation**. GO was synthesized via the modified Hummer's method[25]. One gram of flake graphite and 1 g of sodium nitrate ($NaNO_3$) were mixed in 100 mL of sulfuric acid ($H_2SO_4$, reagent grade, 95–98%) in an ice bath by stirring for 1 h; 6 g of potassium permanganate ($KMnO_4$) was then added and the entire solution was mixed with stirring for 2 h. The reaction proceeded at 35 °C, meanwhile, 46 mL of DI (70 °C) was dripped into the reaction tank slowly. A total of 140 mL of DI water and 20 mL of 30 wt% hydrogen peroxide ($H_2O_2$) were added in the reaction tank and subsequently the synthesis was ended. The GO was washed thoroughly with dilute hydrochloric acid (HCl) and DI water until the GO suspension reached the pH of 7.

To prepare the electrodes, active materials (i.e., GO and PANI), carbon black, and polyvinylidene difluoride (PVDF) were mixed with N-Methyl-2-pyrrolidone (NMP), which were then coated on the carbon papers (CP). The mass ratio of active materials, carbon black, and PVDF was 75:15:10, of which the mass loading of the active materials was around 8–15 mg cm$^{-2}$ after drying at 40 °C for 12 h. Prior to the GO coating, CP was electroplated with Pt nanoparticles by applying cyclic pulse voltage (−1 V for 0.5 s and −1.5 V for 1 s) for 6 min in a commercial Pt electrolyte (Met-Pt 200S, Metalor). PANI (leucoemeraldine base) electrode was doped in 1 M HCl solution for 48 h and converted to protonated leucoemeraldine PANI before cell fabrication. Two electrolytes were used in our experiments: 4 M potassium chloride solution and 0.5 M Iron (II) chloride/0.5 M Iron (III) chloride solution. All chemicals were purchased from Sigma-Aldrich.

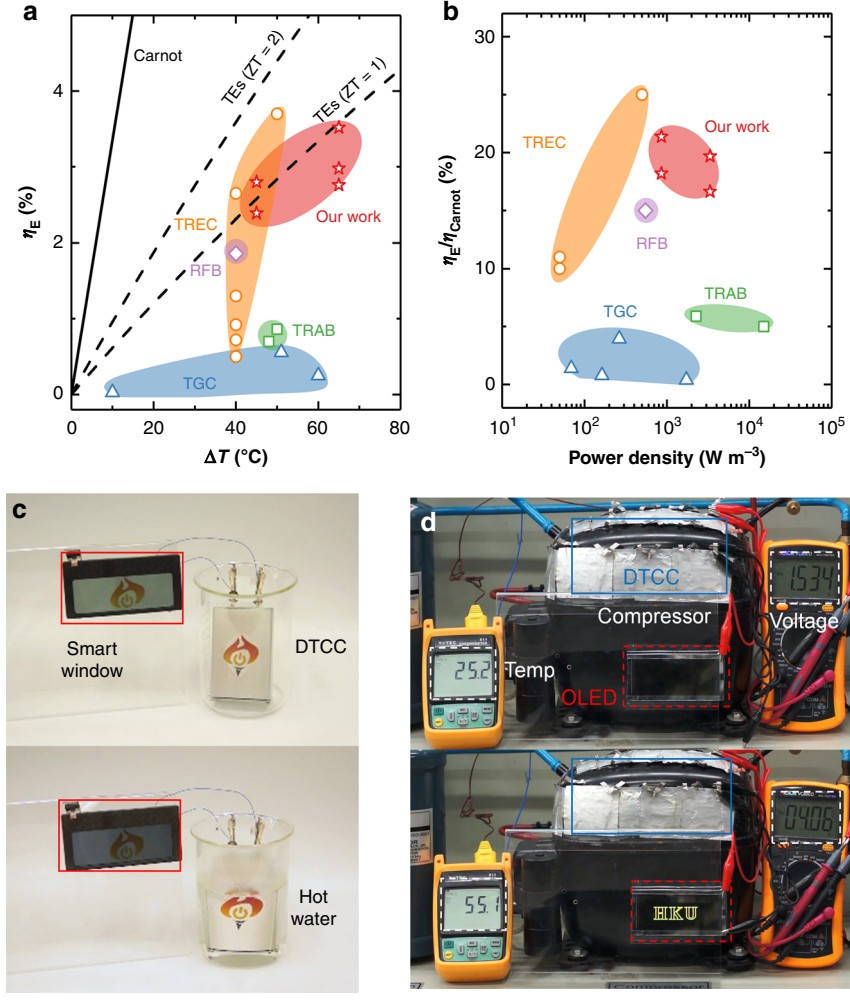

**Fig. 4** Device performance and demonstration. **a** Heat-to-electricity conversion efficiency ($\eta_E$) versus temperature differential, and **b** ratio of $\eta_E$ to $\eta_{Carnot}$ versus volumetric power density for DTCCs and the best-reported TECs including TGCs (blue triangle)[6–9], RFB-based TEC (violet diamond)[14], TRABs (green square)[12,13], and TRECs (orange circle)[10,11,37–39]. **c** Electrochromic smart window powered by the stacked DTCCs in hot water. **d** OLED lighted up by the bendable DTCCs placed along the surface of running compressor

**Pouch cell and temperature controlling system**. DTCC was configured as a pouch cell (Supplementary Fig. 1a), where GO/PtNPs on CP (cathode) was connected to working electrode, and PANI on CP (anode) was connected to counter electrode. The porous hydrophilic separator was located between two electrodes and filled with electrolyte. Titanium foil was used as the reference electrode and the current collector because of its corrosion resistance and stability at high temperature in the electrolyte, and it was connected to a nickel sealant tab for sealing. The potential of Ti foil does not change in our operating temperature range. The entire pouch cell was packaged using aluminum laminated film and the thickness was around 1–1.5 mm. The heating/cooling cycle was carried out by employing two thermoelectric modules, and the temperature was precisely controlled using the Labview program. The setup of temperature controlling system is shown in Supplementary Fig. 1b. Two thermocouples were placed upon and underneath the cell to record the temperatures at both sides. Thermopaste (Omega) was applied to all the interfaces to ensure good thermal contact. The temperature measurement uncertainty was estimated to be ±0.5 °C.

**Material and electrochemical characterization**. Electrochemical tests of the cells were performed using Gamry Reference 3000 Potentiostat. EIS characterization was conducted on a symmetric cell consisting of two identical GO electrodes, which were tested under open circuit condition with the voltage amplitude of 5 mV in the frequency range from $10^{-2}$ to $10^5$ Hz. The anodic onset potential of PANI ($V_{Onset}$) was obtained from the cyclic voltammetry measurement in a three-electrode setup with Ag/AgCl reference electrode and 4 M KCl electrolyte at a scan rate of 50 mV s$^{-1}$. The electrical discharging processes of the cyclability tests were conducted under the constant current of 70 mA g$^{-1}$. The morphology of the electrode materials was characterized by a field-emission scanning electron microscope (Hitachi S-4800). The GO was characterized by a high-resolution transmission electron microscopy (Tecnai, G220S-Twin) and a Raman

spectroscopy (HORIBA, LabRAM HR Evolution). The elemental compositions of GO and PANI were obtained by XPS (Ulvac-PHI, Inc. PHI 5000 VersaProbeII) and Fourier-transform infrared spectroscopy (Perkin-Elmer spectrum-100).

## Data availability
The data that support the plots and findings within this article are available from the corresponding author upon reasonable request.

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

## Acknowledgements

The authors acknowledge constructive discussion with Prof. D. Y. C. Leung and Dr. Y. Chen (The University of Hong Kong), Prof. M.H.K. Leung (City University of Hong Kong), Dr. W. S. Liu (Southern University of Science and Technology), and Mr. Frank H. T. Leung (Techskill (Asia) Limited). The authors also acknowledge Prof. S. W. Liu (Ming Chi University of Technology) for supplying smart window and OLED. The authors acknowledge the financial support of General Research Fund of the Research Grants Council of Hong Kong Special Administrative Region, China, under Award Numbers 17204516 and 17206518, and Innovation and Technology Fund (Ref.: ITS/171/16FX).

## Author contributions

X.W. and S.P.F. developed the concept and designed the experiments. X.W., Y.T.H., K.Y. M., K.H.L., and S.J.W. performed the experiments. X.W., C.L., Y.T.H., and S.P.F. contributed to the interpretation of the results. Y.Y. contributed to the fabrication of pouch cell and measurement system. L.W. contributed to electrode fabrication and design of DTCC's demonstration. C.H.S. participated in the demonstration on smart window and OLED. X.W., C.L., and S.P.F. co-wrote the manuscript.

## Additional information

**Competing interests:** A provisional patent in the title of "Thermo-electric capacitor" (U. S. 62/617514, PCT/CN2019/071777) has been filed by the applicant of Versitech Limited (Technology Transfer Office, the University of Hong Kong) with the inventors Shien-Ping Feng, Xun Wang, Yu-Ting Huang, Zeyang Zheng, Lei Wang, Ka Ho Li, and Kaiyu Mu.

