## [Peer Review File · Nature Communications]

Reviewers' comments:

Reviewer #1 (Remarks to the Author):

In this paper, the authors reported a method to convert heat into electricity by a combination of a supercapacitor electrode (oxidized graphite) and a battery electrode (PANI). It was observed that when the cell is heated, a large voltage is generated. The cell can be discharged at high temperature, and regenerated when it cools down. KCl and Fe-Cl electrolytes are tested and it was observed the use of Fe-Cl electrolyte significantly increases the capacity.

This is a very interesting paper and opens new grounds in converting low-grade waste heat into electricity. The efficiency achieved $\sim 5.8\%$ at $70\text{ }^{\circ}\text{C}$ without considering heat recuperation, is very impressive and probably a world record for converting low-grade heat into electricity. The cell regeneration when cooled down makes it attractive for many applications. The energy conversion cycle described is new in principle. The authors have carried out extensive studies to understand the mechanisms involved, although there are many questions remain. However, based on its potential technological impacts, I strongly recommend the acceptance of the paper. Below are minor comments for the authors to consider when submitting a revised manuscript.

1. Have authors tested Go/PtNP/KCl/PANI cell? The paper has not described clearly how the Pt catalyst work.
2. It will be nice to show the traces of a few cycles on same diagram. Fig. 11 in SI does show them, I also would recommend a figure overlaying them. I am curious to see if VOC returns to same value after cooling down.
3. The paper says reaction mechanisms change above $50\text{ }^{\circ}\text{C}$. Why alpha seems to be linear over the entire temperature range? One explanation is that the data points in Fig.2 is not dense. One could see increase in alpha between $30\text{-}50$ and $50\text{-}70$ for PtNP sample is higher. More points will settle this question.
4. SI should include how EDL contribution is calculated.
5. Why initial discharge is so fast? (Fig.2c)
6. P.5, the claim of endothermic reaction seems to be unsubstantiated. Is it necessary?

Minor comments:

7. Separate equation number in SI from equation
8. Comment on Ref. 10 focusing on electrical charging only is not complete, since non-electrical charging TREC also was developed.
9. Alpha is not defined in the main text when first cited (p.3)
10. P.5, is the claim of endothermic reaction
11. P.13, "In prior to" should be "Prior to".

Reviewer #2 (Remarks to the Author):

The authors reported a novel thermal cell that can be directly charged by waste heat and delivers electrical current. The thermal cell is self-regenerative when cooled down. The asymmetric electrodes scheme and the working mechanism presented here is opening the way for alternative and more efficient way to utilize waste heat. However, the reviewer don't think the authors present clearly on the mechanism for working and on how to calculate the efficiency of the system. Some comments from the reviewer are as follows:

1. The introduction could be improved. The authors simply listed waste heat energy harvesting devices previous reported and compared their performance in terms of the conversion efficiency. However, the authors should emphasize the features of the proposed DTCC system by comparing the advantages and disadvantages of each device more systematically. In particular, it is necessary to explain how it is different in comparison with the previously reported "charging free electrochemical system (PNAS

(2014), 111, 17011-17016”.

2. The authors should explain the proposed working mechanism more clearly in Figure 1.

2-1. When the temperature of the DTCC system is raised from TL to TH, is there any change in the redox potential of PANI electrode depending on the temperature? According to Figure 3(a4), this effect should be reflected in the schematic for working mechanism.

2-2. For the schematic in Figure 1(b2), the authors said “the reduction reaction of Fe^{3+} happens ...” Can the voltage due to the so-called ionic Seebeck effect be produced at the two electrodes placed at the same temperature (TH) without the use of cationic or anionic membrane? In other words, is it possible that only a half-cell reaction occurs at the isothermal condition? The reviewer believes that there is no reduction of Fe^{3+} but the voltage can be developed due to the potential between the electrode potential and the LUMO level of the ion.

2-3. Is the output voltage delivered from the cell equal to the sum of voltages from GO pseudocapacitive and the thermogalvanic effects? And how about the total current from the cell?

2-4. The cell voltage shown in Figure 3(a) during thermal charging should be “VGO/PtNPs – VPANI”, but the peak cell voltage is higher than the VGO/PtNPs.

3. The total cell voltage should be the “VGO/PtNPs – VPANI”, but, as shown in Figure 3(a2), the measured cell voltage is higher than the VGO/PtNPs. That should be clearly explained in conjunction with the working mechanism shown in Figure 1.

4. On page 9, the performance of the DTCC has been reduced by 20% after 20 cycles, so it would be good to give specific reasons for the performance degradation.

5. In calculation of the efficiency of DTCC system, the output electrical work (W in Table 1 of SI) might be calculated using the total underlying area of the voltage curve in Figure 3(a2). Did the authors exclude the work area covered by the “built-in” V_{oc} for that electrical work calculation? If not, the work area should be excluded as considering the cycle for charging/discharging.

Reviewer #3 (Remarks to the Author):

This manuscript claims the major device is "direct thermal charging cell", which is more like the "thermal capacitor" device due to charging-discharging process of active materials at both anode and cathode. This could raise a critical question on the real mechanism of the device performance. As the whole performance will be the combination of "Thermal Electronic Devices and Thermal capacitors", which is good but has to be clarified.

In addition, in Figure S2, it shows a shoulder of potential increase to reach the stable platform, which means the active materials (Go and/or PANi) has took time to changes their states due to the potential difference. Even though the authors claimed that the device could be regenerated, no real cycling performance were provided with proper materials and electrode characterization before and after cycling.

Therefore, I would suggest the authors to conduct a proper major revision of current manuscript before it is considered for publication.

Revision/Rebuttal Report for NCOMMS-19-102626-T

Direct Thermal Charging Cell for Converting Low-grade Heat to Electricity

Xun Wang, Yu-Ting Huang, Chang Liu, Kaiyu Mu, Ka Ho Li, Sijia Wang, Yuan Yang, Lei Wang, Chia-Hung Su, Shien-Ping Feng

We thank the editor for handling the manuscript review and the reviewers for their detailed and insightful comments. The manuscript has been revised to reflect the comments of editor and all reviewers. A detailed revision/rebuttal report is included below, and all revised text in the manuscript and supplementary material is marked in blue for convenience.

Reviewer #1

In this paper, the authors reported a method to convert heat into electricity by a combination of a supercapacitor electrode (oxidized graphite) and a battery electrode (PANI). It was observed that when the cell is heated, a large voltage is generated. The cell can be discharged at high temperature, and regenerated when it cools down. KCl and Fe-Cl electrolytes are tested and it was observed the use of Fe-Cl electrolyte significantly increases the capacity.

This is a very interesting paper and opens new grounds in converting low-grade waste heat into electricity. The efficiency achieved ~5.8% at 70 °C without considering heat recuperation, is very impressive and probably a world record for converting low-grade heat into electricity. The cell regeneration when cooled down makes it attractive for many applications. The energy conversion cycle described is new in principle. The authors have carried out extensive studies to understand the mechanisms involved, although there are many questions remain. However, based on its potential technological impacts, I strongly recommend the acceptance of the paper. Below are minor comments for the authors to consider when submitting a revised manuscript.

1. Have authors tested GO/PtNP/KCl/PANI cell? The paper has not described clearly how the Pt catalyst work.

Response: We appreciate the reviewer's positive comment that "*I strongly recommend the acceptance of the paper*". As suggested by the reviewer, we add the thermal charging response of GO/PtNP|KCl|PANI cell in Fig. 2a and Supplementary Fig. 2c, of which the temperature coefficient is 3.1 mV/K. Therefore, the three cells (GO|KCl|Ti, GO|KCl|PANI, GO/PtNPs|KCl|PANI), have the similar α of 3.5 mV/K, 3.3 mV/K and 3.1 mV/K, respectively. With the aid of Fe²⁺/Fe³⁺, GO/PtNPs|Fe²⁺/Fe³⁺|PANI DTCC obtains the higher voltage compared to those of three cells and reaches a α of 5.0 mV/K. The results indicate the decisive role of GO and little contribution of PANI to voltage generation and the occurrence of thermogalvanic effect between Fe²⁺/Fe³⁺ and GO/PtNPs for further increased V_{OC}. Here we duplicate the revised Fig. 2a and Supplementary Fig. 2 below as Fig. R1.1 and R1.2 to make it convenient for the reviewer to

read. The related sentences have been revised correspondingly in the revised manuscript (highlighted in blue in line 132 on page 6).

Fig. R1.1| V_{OC} versus temperature for four cells, GO|KCl|Ti, GO|KCl|PANI, GO/PtNPs|KCl|PANI and GO/PtNPs|Fe²⁺/Fe³⁺|PANI.

Fig. R1.2| Open circuit voltages of **a**, GO|KCl|Ti cell, **b**, GO|KCl|PANI cell, **c**, GO/PtNPs|KCl|PANI cell and **d**, GO/PtNPs| Fe²⁺/Fe³⁺|PANI DTCC measured at 50 °C, 70 °C and 90 °C; **e**, Measured temperature profiles of cells.

PtNPs are vital in both thermal-charging and electrical discharging process. In thermal-charging process, the reduction reaction of Fe^{3+} happens under the catalysis of GO/PtNPs at T_H due to the positive α of $\text{Fe}^{2+}/\text{Fe}^{3+}$, which further enhances the V_{OC} . In the discharging process, with the aid of catalytic PtNPs, the Fe^{3+} effectively carries the electrons from GO/PtNPs cathode and is reduced to Fe^{2+} , which prevents the oxygen functional groups of GO from being reduced and consumed to retain the thermal-induced voltage and thus sustains the oxidation of PANI. Here, we add experiments to better understand the role of Pt catalyst in DTCC system. Fig. R1.3 is the cyclic voltammetry (CV) for GO-coated carbon paper (GO-CP) and Pt-coated CP (Pt-CP) in the electrolyte of 0.5M $\text{FeCl}_2/\text{FeCl}_3$ at 50°C. As compared with GO, PtNPs significantly enhance the catalytic ability for $\text{Fe}^{2+/3+}$ redox reactions.

Moreover, in another experiment, the GO/PtNPs electrode was immersed in 0.5M FeCl_3 solution (Fe^{3+} -only electrolyte) and then heated to 50°C for 2 min. Potassium ferricyanide solution ($\text{K}_3[\text{Fe}(\text{CN})_6]$) can be used to detect Fe^{2+} in the solution. After adding few drops of 0.05M $\text{K}_3[\text{Fe}(\text{CN})_6]$ solution into the electrolyte with GO/PtNPs, the blue precipitate was formed and dispersed in the electrolyte so that the solution became a green color (Fig. R1.4), evidencing the existence of Fe^{2+} ions in the solution as Fe^{2+} ions would react with ferricyanide ions to form Turnbull's blue ($\text{KFe}[\text{Fe}(\text{CN})_6]$) particles¹. The control experiment (0.5M FeCl_3 solution without GO/PtNPs was heated to 50°C for 2 min) shows a brownish color due to the addition of $\text{K}_3[\text{Fe}(\text{CN})_6]$ solution. The above experimental result provides evidence that GO/PtNPs would indeed catalyze the reduction reaction of Fe^{3+} to Fe^{2+} when heating in open circuit condition. The possible chemical reaction is shown as below,

The abovementioned results have been newly added as Supplementary Note 6, and Fig. R1.3 and Fig. R1.4 are added as Supplementary Fig. 8 and 9, respectively.

Fig. R1.3 Cyclic voltammetry of Pt-CP and GO-CP in 0.5M $\text{FeCl}_2/\text{FeCl}_3$ electrolyte at 50°C. The scan rate is 50 mV/s.

Fig. R1.4 | **a**, 0.5M FeCl_3 solution with and without GO/PtNPs electrode after heating; **b**, after adding 0.05M $\text{K}_3[\text{Fe}(\text{CN})_6]$ solution, the sample without GO/PtNPs shows a brownish color while that with GO/PtNPs turns to green color.

2. It will be nice to show the traces of a few cycles on same diagram. Fig. 11 in SI does show them, I also would recommend a figure overlaying them. I am curious to see if V_{OC} returns to same value after cooling down.

Response: We thank the reviewer for the suggestion. We have added the overlapping of thermal-charging/discharging curves in Supplementary Fig.16b. As seen, the V_{OC} has a slight shift after the first cycle and keeps the same value in the following cycles. Basically, the reproducible thermal-induced voltage and current are obtained during heating/cooling cycles. Here we duplicate the Supplementary Fig.16 below as Fig. R1.5 to make it convenient for the reviewer to read.

Fig. R1.5 | **a**, Cell voltage and current *versus* time during intermittent heating/cooling cycles; **b**, Overlapping of voltage-time curves for different cycles.

3. The paper says reaction mechanisms change above 50 oC. Why alpha seems to be linear over the entire temperature range? One explanation is that the data points in Fig.2 is not dense. One could see increase in alpha between 30-50 and 50-70 for PtNP sample is higher. More points will settle this question.

Response: We thank the reviewer for his/her good observation and suggestion. The data points with only four testing temperatures in Fig. 2a are not dense enough; the testing temperatures in the experiment of electrochemical impedance spectroscopy (EIS) are not dense as well. Therefore, we have done experiments at different temperatures (30°C, 40 °C, 50 °C, 60 °C, 70 °C, 90 °C) to add more data points in Fig. 2a (cell V_{OC} versus temperature) and Supplementary Fig. 3 (EIS). With more data points, we found that the voltage generated by temperature-induced pseudocapacitive effect is quite linear with temperature in the testing temperature range from 30 °C to 90 °C. There is no obvious turning point or starting temperature. In conclusion, once the temperature is raised, a fast and reversible faradaic process would be induced by the chemisorption of protons on the oxygen functional groups of the GO.

In the manuscript, it is noted that the operating temperature of DTCCs is in the range of 40-90°C, as the boiling point of the aqueous electrolyte determines the upper-temperature limit. The lower-temperature limit is set to 40°C because the current can only be produced when $V_{OC(Cell)} > V_{Onset(PANI)}$ so that PANI is oxidized to release electrons (Fig. 2d). Hence, the operating temperature of DTCCs is determined by the boiling point of electrolyte and the oxidation of PANI.

Therefore, we revised the manuscript (highlighted in blue in line 146 on page 7) and EIS results in supplementary information correspondingly (Supplementary Note 4). The revised Fig. 2a has been shown as Fig R1.1 in Question 1. Here we duplicate the revised Supplementary Fig. 3 below as R1.6 to make it convenient for the reviewer to read.

Fig. R1.6 | EIS of GO|KCl|GO symmetric cell at different temperatures. **a**, Nyquist plot (insert: magnified figure in high-frequency region). Complex capacitance plots: **b**, real capacitance and **c**, imaginary capacitance versus frequency.

4. *SI should include how EDL contribution is calculated.*

Response: We thank the reviewer for his/her comment. The calculation of EDL contribution has been added as Supplementary Note 3.

5. *Why initial discharge is so fast? (Fig.2c)*

Response: In our current setup, it is believed that the mechanical contact between electrode and current collector causes a relatively high contact resistance, leading to a fast IR drop at the beginning of discharging process. We are now working on the electrodeposition of thin SnAg film (as solder joint) on the backside of electrode in order to improve the contact performance between electrode and current collector, which will increase the power output of DTCC. We will publish the results in the future.

6. *P.5, the claim of endothermic reaction seems to be unsubstantiated. Is it necessary?*

Response: We agree the reviewer's comment. The claim of endothermic reaction is not necessary. We have removed the sentence in the revised manuscript.

Minor comments:

7. *Separate equation number in SI from equation*

Response: We thank the reviewer for his/her comment. The equation numbers have been separated from equations in SI.

8. *Comment on Ref. 10 focusing on electrical charging only is not complete, since non-electrical charging TREC also was developed.*

Response: We thank the reviewer for pointing this out. The charging-free TREC has been included in the introduction (highlighted in blue in line 55 on page 3). Moreover, as suggested by reviewer #2 and #3, we also add Supplementary Table 1 to compare the features for the previous reported TEC systems and our DTCC.

9. *Alpha is not defined in the main text when first cited (p.3)*

Response: We thank the reviewer for his/her comment. The definition of α has been added in the revised manuscript. (highlighted in blue in line 39 on page 2).

10. P.5, *is the claim of endothermic reaction*

Response: We thank the reviewer for his/her comment. The claim of endothermic reaction is not necessary. We have removed the sentence in the revised manuscript.

11. P.13, *“In prior to” should be “Prior to”*.

Response: We thank the reviewer for his/her comment. The sentence has been revised accordingly.

Reviewer #2

The authors reported a novel thermal cell that can be directly charged by waste heat and delivers electrical current. The thermal cell is self-regenerative when cooled down. The asymmetric electrodes scheme and the working mechanism presented here is opening the way for alternative and more efficient way to utilize waste heat. However, the reviewer don't think the authors present clearly on the mechanism for working and on how to calculate the efficiency of the system. Some comments from the reviewer are as follows:

1. The introduction could be improved. The authors simply listed waste heat energy harvesting devices previous reported and compared their performance in terms of the conversion efficiency. However, the authors should emphasize the features of the proposed DTCC system by comparing the advantages and disadvantages of each device more systematically. In particular, it is necessary to explain how it is different in comparison with the previously reported “charging free electrochemical system (PNAS (2014), 111, 17011-17016)”.

Response: We thank the reviewer for his/her comment. We revised the introduction with more details in advantages/disadvantages for different TEC technologies, and particularly added a Supplementary Table 1 to compare the features for the reported TEC systems and our DTCC. As mentioned in the revised manuscript, although charging-free TREC (PNAS (2014), 111, 17011-17016) was demonstrated, the low efficiency of <1% (without heat recuperation) and the use of expensive ion-selective membrane are concerns. The voltages of charging-free TREC are opposite at low and high temperatures. Therefore, the two electrochemical processes at both low and high temperatures in a cycle can be discharged by current flow in reverse direction. This work requires a temperature cycle to complete charge and discharge.

DTCC utilizes the thermo-pseudocapacitive effect to build a higher voltage and employs the redox couple in electrolyte to carry electrons. The charge and discharge are both operated under isothermal heating. The cell is self-regenerated when cooled down without electron flow. Therefore, the energy conversion cycle with isothermal charging/discharging operation at high temperature and regenerative mode at low temperature is different from the charging-free TREC and all other existing TEC technologies. Here we duplicate the newly added Supplementary Table 1 below as Table R2.1 to make it convenient for the reviewer to read. The related sentences have been revised correspondingly in the revised manuscript (highlighted in blue on page 3).

Table R2.1| Comparison of different TEC systems for low grade heat to electricity conversion

Operating mode	TEC system	Structure and Materials	α (mV/K)	η_E (η_E/η_{Carnot})	Advantages & Limitations	Ref
Temperature gradient (continuous operation based on temperature-dependent redox potentials at the hot and cold sides)	TGC	Electrode: Multi-walled carbon nanotubes (MWCNT) based electrode Electrolyte: $K_3Fe(CN)_6/K_4Fe(CN)_6$	1.4	0.24% (1.4%)	Advantage: efficiency can be improved by using MWCNT instead of Pt electrodes. Limitation: low efficiency caused by low electrolyte conductance	1
		Electrode: Carbon-based material Electrolyte: $K_3[Fe(CN)_6]/(NH_4)_4[Fe(CN)_6]$ or $Fe_2(SO_4)_3/FeSO_4$	1.85	0.11% (0.4%)	Advantage: high power density by using improved electrolyte, electrolyte filled thermal separator and optimized carbon electrode materials. Limitation: low efficiency	2
		Electrode: CNT aerogel sheets Electrolyte: $K_3Fe(CN)_6/K_4Fe(CN)_6$	1.43	0.55% (3.95%)	Advantage: relative high efficiency and power density in TGCs Limitation: low efficiency	3
Temperature gradient (flowing electrolytes in symmetric redox reactions at different temperatures)	RFB	Electrode: carbon cloth Flow electrolyte: $[Fe(CN)_6]^{3-}/[Fe(CN)_6]^{4-}$ and V^{3+}/V^{2+}	3.0	1.8% (15%)	Advantage: relative high efficiency, high α , continuous operation across a broad range of temperatures Limitation: relative high cost in electrolyte and operation	4
Temperature cycle (operating between hot and cold reservoirs alternating in a thermal cycle; charging and discharging at different temperatures)	TREC	Electrode: CuHCF and Cu Electrolyte: $NaNO_3$ and $Cu(NO_3)_2$	1.2	3.7% (25%)	Advantages: high efficiency Limitation: the need of external electricity for charging process and the use of ionic selective membrane	5
		Electrode: NiHCF and Ag/AgCl Electrolyte: KCl	0.74	1.6% (13%)	Advantages: simple pouch cell structure without using ionic selective membrane Limitation: the need of external electricity for charging process, relative low efficiency	6
		Electrode: $KFe^{II}Fe^{III}(CN)_6$ and $K_3Fe(CN)_6/K_4Fe(CN)_6$ with carbon cloth Electrolyte: KNO_3	1.45	0.72% (6.0%)	Advantages: charging-free system Limitation: low efficiency, the use of ionic selective membrane	7
	TRAB	Electrode: Cu Electrolyte: $Cu(NO_3)_2/NH_4NO_3$	–	0.86% (6.1%)	Advantages: high power density, low cost Limitation: ammonia stream is a serious concern regarding the leakage, stability, and safety	8
		Electrode: Cu Flow electrolyte: $Cu(NO_3)_2/NH_4NO_3$ flow	–	0.70% (5.0%)	Advantages: high power density, continuous operation Limitation: ammonia stream is a serious concern regarding the leakage, stability, and safety	9
Temperature cycle (thermal-charging and discharging under high temperature and self-regenerated at low temperature)	DTCC	Electrode: GO/PtNPs and PANI Electrolyte: $FeCl_2/FeCl_3$	5.0	5.84% (44.5%)	Advantages: both high efficiency and power density, high α , low cost and easy operation mode, simple pouch cell structure without using ionic selective membrane Limitation: the degradation of long-term cycling	This work

2. The authors should explain the proposed working mechanism more clearly in Figure 1.

2-1. When the temperature of the DTCC system is raised from T_L to T_H , is there any change in the redox potential of PANI electrode depending on the temperature? According to Figure 3(a4), this effect should be reflected in the schematic for working mechanism.

Response: We thank the reviewer to point this out. According to our experiment as shown in Fig. 3a4, the redox potential of PANI has a little response to temperature with a small positive temperature coefficient of < 0.2 mV/K. We agree the reviewer that this effect should be reflected in the schematic for working mechanism in Fig. 1b. Here we duplicate the revised Fig. 1b below as Fig. R2.1 to make it convenient for the reviewer to read. The related sentences have also been revised correspondingly in the revised manuscript (highlighted in blue in line 91 on page 4 and line 201 page 9).

Fig. R2.1 | **b₀**, Built-in voltage (ΔV_0) based on the differential of electrochemical potentials between two electrodes at T_L . Heat-to-electricity conversion undergoes three stages. Stage 1: thermal charging process, DTCC is heated up from T_L to T_H in open circuit condition; **b₁**, pseudocapacitive reactions occur at the GO-aqueous interface, generating a thermal-induced voltage, and **b₂**, the V_{oc} is further enlarged with the simultaneous reduction reaction of Fe^{3+} at T_H . Stage 2: **b₃**, electrical discharging process, DTCC produces current via the oxidation of PANI and the reduction of Fe^{3+} at T_H . Stage 3: **b₄**, self-regeneration process, PANI and Fe^{3+} are chemically regenerated at T_L .

2-2. For the schematic in Figure 1(b2), the authors said “the reduction reaction of Fe³⁺ happens ...” Can the voltage due to the so-called ionic Seebeck effect be produced at the two electrodes placed at the same temperature (T_H) without the use of cationic or anionic membrane? In other words, is it possible that only a half-cell reaction occurs at the isothermal condition? The reviewer believes that there is no reduction of Fe³⁺ but the voltage can be developed due to the potential between the electrode potential and the LUMO level of the ion.

Response: We thank the reviewer for his/her comment. In Fig. 1b₂, we mentioned that the reduction reaction of Fe³⁺ happens under the catalysis of GO/PtNPs at T_H due to the positive α of Fe²⁺/Fe³⁺, which further enhances the V_{OC}. Actually, our DTCC generates voltage via temperature-induced pseudocapacitive effect of GO together with thermogalvanic effect of Fe²⁺/Fe³⁺ (catalyzed by GO/PtNPs) when heating in open circuit condition. It is different from ionic Seebeck effect by using cationic or anionic membrane.

To better understand the catalysis of GO/PtNPs for the reduction reaction of Fe³⁺ at T_H in DTCC system, we have done more experiments. The GO/PtNPs electrode was immersed in 0.5M FeCl₃ solution (Fe³⁺-only electrolyte) and then heated to 50°C for 2 min. Potassium ferricyanide solution (K₃[Fe(CN)₆]) can be used to detect Fe²⁺ in the solution. After adding few drops of 0.05M K₃[Fe(CN)₆] solution into the electrolyte with GO/PtNPs, the blue precipitate was formed and dispersed in the electrolyte so that the solution became a green color (Fig. R2.2), evidencing the existence of Fe²⁺ ions in the solution as Fe²⁺ ions would react with ferricyanide ions to form Turnbull’s blue (KFe[Fe(CN)₆]) particles¹. The control experiment (0.5M FeCl₃ solution without GO/PtNPs was heated to 50°C for 2 min) shows a brownish color due to the addition of K₃[Fe(CN)₆] solution. The above experimental result provides evidence that GO/PtNPs would indeed catalyze the reduction reaction of Fe³⁺ to Fe²⁺ when heating in open circuit condition. It is noted that we may exclude the effect of PANI for the redox reaction of Fe^{2+/3+} during heating because the potential of PANI anode (V_{PANI}) has very little voltage change (Fig. 3a₄).

Here, we agree the reviewer that it is not only a half-reaction occurring in the system. The full chemical reaction at GO/PtNPs side when heating in open circuit condition is shown as below,

We also thank the reviewer to describe the point of the lowest unoccupied molecular orbital (LUMO) of Fe³⁺. As the above chemical reaction, the electrons transfer from the GO/PtNPs to LUMO of Fe³⁺ during heating conduces to an enlarged V_{OC}. The theoretical LUMO level of Fe³⁺ (−8.74 eV) is lower than the Fermi level of GO (−5.0 eV), making it easier to receive electrons from GO/PtNPs electrode to obtain a larger electrochemical potential difference between two electrodes. The abovementioned results have been newly added as Supplementary Note 6 and Supplementary Fig. 9, respectively. The related sentences have been revised correspondingly in the revised manuscript (highlighted in blue in line 178 on page 8).

Fig. R2.2 | **a**, 0.5M FeCl_3 solution with and without GO/PtNPs electrode after heating; **b**, after adding 0.05M $\text{K}_3[\text{Fe}(\text{CN})_6]$ solution, the sample without GO/PtNPs shows a brownish color while that with GO/PtNPs turns to green color.

2-3. *Is the output voltage delivered from the cell equal to the sum of voltages from GO pseudocapacitive and the thermogalvanic effects? And how about the total current from the cell?*

Response: We thank the reviewer for his/her comment. Yes, the voltage is generated via temperature-induced pseudocapacitive effect of GO and thermogalvanic effect of $\text{Fe}^{3+}/\text{Fe}^{2+}$. As shown in Fig. 2a, three cells (GO|KCl|Ti, GO|KCl|PANI, GO/PtNPs|KCl|PANI) have the similar α of 3.5 mV/K, 3.3 mV/K and 3.1 mV/K, respectively, while the cell of GO/PtNPs| $\text{Fe}^{2+/3+}$ |PANI reaches a markedly high α of 5.0 mV/K. The results indicate the decisive role of GO and little contribution of PANI to voltage generation and the occurrence of thermogalvanic effect between $\text{Fe}^{2+}/\text{Fe}^{3+}$ and GO/PtNPs for further increased V_{OC} . In Fig. 2a, the cell V_{OC} is about 390 mV when heated up to 90°C, of which α is 5.0 mV/K [$\alpha = (394-96) \text{ mV}/(90-30) \text{ }^\circ\text{C}$]. Therefore, the V_{OC} should be the sum of voltages from GO pseudocapacitive and the thermogalvanic effects (we add a sentence to describe this part more clearly in Supplementary Note 2). However, there is a fast IR drop (voltage drop) at the beginning of discharging process because the mechanical contact between electrode and current collector causes a relatively high contact resistance. We are now working on the electrodeposition of thin SnAg film (as solder joint) on the backside of electrode in order to improve the contact performance between electrode and current collector.

When an external load is connected, the DTCC is discharged continuously at T_{H} because the oxidation of PANI anode and the simultaneous reduction of Fe^{3+} on catalytic GO/PtNPs cathode. In Fig. 2e, to compare with the control cell without using $\text{Fe}^{2+}/\text{Fe}^{3+}$, the total current output of DTCCs is mainly dominated by the reduction of Fe^{3+} to Fe^{2+} (the majority of electrons from PANI are carried by the reduction reaction of Fe^{3+} to Fe^{2+} in cathode side). This part has been mentioned in the manuscript in line 181 on page 8.

2-4. The cell voltage shown in Figure 3(a) during thermal charging should be “ $V_{GO/PtNPs} - V_{PANI}$ ”, but the peak cell voltage is higher than the $V_{GO/PtNPs}$.

Response: We thank the reviewer for his/her good observation. It is correct that the total V_{OC} equals to “ $V_{GO/PtNPs} - V_{PANI}$ ”. Here, the peak cell voltage is higher than the $V_{GO/PtNPs}$ because V_{PANI} shows a negative voltage versus Ti reference electrode at around 150s, causing $V_{oc} = V_{GO/PtNPs} - V_{PANI} = 372 \text{ mV} - (-31 \text{ mV}) = 403 \text{ mV} > V_{GO/PtNPs} = 372 \text{ mV}$. We amplified the voltage profiles in the thermal charging process of Fig. 3a₂-a₄, as shown in Fig. R2.3 below.

Fig. R2.3| Voltage profiles of V_{OC} , $V_{GO/PtNPs}$ and V_{PANI} in thermal charging process.

3. The total cell voltage should be the “ $V_{GO/PtNPs} - V_{PANI}$ ”, but, as shown in Figure 3(a₂), the measured cell voltage is higher than the $V_{GO/PtNPs}$. That should be clearly explained in conjunction with the working mechanism shown in Figure 1.

Response: We thank the reviewer for his/her good observation. It has been explained in question 2-4.

4. On page 9, the performance of the DTCC has been reduced by 20% after 20 cycles, so it would be good to give specific reasons for the performance degradation.

Response: We thank the reviewer for his/her suggestion. The performance of DTCC was degraded about 20% after 20 cycles. Here, we add an experiment of Fourier-transform infrared spectroscopy (FTIR) to characterize materials and electrodes to understand the performance degradation. Fig. R2.4 shows the FTIR spectra of GO/PtNPs electrodes and PANI electrodes before cycling, after 10 cycles and after 20 cycles. GO/PtNPs electrodes and PANI electrodes remained unchanged in the first 10 cycles, but peaks were changed and shifted after 20 cycles. As described in XPS results of Supplementary Note 7, PANI is oxidized during electrical discharging and refreshed at room temperature via self-regeneration, which can also be confirmed by the unchanged FTIR features after 10 cycle. The IR bands at 1568 cm^{-1} and 1494 cm^{-1} are assigned to C=C stretching vibrations of benzenoid and aromatic ring while that at 1634 cm^{-1} is attributed to C=N bond and C=C bond vibration². After 20 cycles, the shift from 1568 cm^{-1} to 1634 cm^{-1} corresponds to a transformation

from amine link to imine link of PANI, indicating the oxidation of PANI. Therefore, the PANI is still gradually oxidized and consumed after a long-term cycling. The peaks of GO become smaller due to the consumption of oxygen functional groups or the materials peeled off during cycles. One possible solution is to take the cell apart after tens of cycles and reassemble it after regenerating electrodes via chemical treatments^{3,4}.

Other than the problem of material degradation, in the current set-up, the cycling stability is also limited by the aqueous electrolyte, of which the large volume expansion and shrinkage during temperature cycles cause the active materials peeled off and the sealing problem (Fig. R2.5). The improvement of the cyclability of DTCCs can be expected with further optimization of electrode, electrolyte, and cell packaging.

The abovementioned results have been newly added as Supplementary Note 8 with Supplementary Fig. 13 and 14. The related sentences have been revised correspondingly in the revised manuscript (highlighted in blue in line 212 on page 10).

Fig. R2.4 FTIR spectra of **a**, GO/PtNPs electrodes and **b**, PANI electrodes before cycling, after 10 cycles and after 20 cycles.

Fig. R2.5 Observed problems in current set-up: **a**, leakage of electrolyte and **b**, formation of cracks on GO and PANI electrode.

5. In calculation of the efficiency of DTCC system, the output electrical work (W in Table 1 of SI) might be calculated using the total underlying area of the voltage curve in Figure3(a2). Did the authors exclude the work area covered by the “built-in” V_{oc} for that electrical work calculation? If not, the work area should be excluded as considering the cycle for charging/discharging.

Response: We thank the reviewer for his/her comment. We stopped the discharging at the built-in ΔV_0 and the output energy with the voltage lower than ΔV_0 was not included. Take Fig. 3a2 as an example (Fig. R2.6), the total cell capacitance (C) was calculated by using the linear part of discharging curve. Then, the effective output energy (W_{eff}) can be calculated by

$$W_{eff} = \frac{1}{2}CV_i^2 - \frac{1}{2}C(\Delta V_0)^2$$

V_i is used instead of V_{oc} due to the IR drop. Here, the $\frac{1}{2}C(\Delta V_0)^2$ represents the work covered by the built-in voltage, which was actually subtracted in the calculation. The average efficiency reported in this work was calculated by using multiple cells. The related description is revised to express it more clearly in Supplementary Note 10 (highlighted in blue in line 313 on page 25 in supplementary information).

Fig. R2.6 | Discharging profiles and efficiency calculations details.

Reviewer #3

1. This manuscript claims the major device is "direct thermal charging cell", which is more like the "thermal capacitor" device due to charging-discharging process of active materials at both anode and cathode. This could raise a critical question on the real mechanism of the device performance. As the whole performance will be the combination of "Thermal Electronic Devices and Thermal capacitors", which is good but has to be clarified.

Response: We thank the reviewer for his/her comment. In this work, we focus on liquid-based thermoelectrochemical cells (TECs) instead of solid-state thermoelectric devices (TEs) because liquid-based TECs is promising for low-grade heat harvesting as their temperature coefficient (α) is one order of magnitude higher than those of TEs (Seebeck coefficient of 100-200 $\mu\text{V/K}$). Considerable attempts to design TECs is categorized into two means of creating temperature gradient: temperature gradient in cell configuration (spatial-scale) and temperature difference in thermal cycle (time-scale). For example, TGC is continuously operated under a temperature gradient based on temperature-dependent redox potentials at the hot and cold sides. TREC and TRAB is operated between hot and cold reservoirs that alternates in a thermal cycle, where cells are charged and discharged at different temperatures.

The operating mode is different between the previous reported TECs and our DTCC. DTCC utilizes the thermopseudocapacitive effect to build a higher voltage and employs the redox couple in electrolyte to carry electrons (the detailed mechanism can be referred to Fig. 1b in the manuscript). The charge and discharge are both operated under isothermal heating (high temperature). The cell is self-regenerated when cooled down (low temperature) without electron flow. Hence, the energy conversion cycle with isothermal charging/discharging operation at high temperature and regenerative mode at low temperature is fundamentally different from TE, TGC, TRAB, and TREC (including charging-free TREC).

We revised the introduction with more details in advantages/disadvantages for different TEC technologies, and particularly added a Supplementary Table 1 to compare the features for the reported TEC systems and our DTCC. Here we duplicate the newly added Supplementary Table 1 below as Table R3.1 to make it convenient for the reviewer to read. The related sentences have been revised correspondingly in the revised manuscript (highlighted in blue in page 2 and page 3).

Table R3.1| Comparison of different TEC technologies for low grade heat to electricity conversion

Operating mode	TEC system	Structure and Materials	α (mV/K)	η_E (η_E/η_{Carnot})	Advantages & Limitations	Ref
Temperature gradient (continuous operation based on temperature-dependent redox potentials at the hot and cold sides)	TGC	Electrode: Multi-walled carbon nanotubes(MWCNT) based electrode Electrolyte: $K_3Fe(CN)_6/K_4Fe(CN)_6$	1.4	0.24% (1.4%)	Advantage: efficiency can be improved by using MWCNT instead of Pt electrodes. Limitation: low efficiency caused by low electrolyte conductance	1
		Electrode: Carbon-based material Electrolyte: $K_3[Fe(CN)_6]/(NH_4)_4[Fe(CN)_6]$ or $Fe_2(SO_4)_3/FeSO_4$	1.85	0.11% (0.4%)	Advantage: high power density by using improved electrolyte, electrolyte filled thermal separator and optimized carbon electrode materials. Limitation: low efficiency	2
		Electrode: CNT aerogel sheets Electrolyte: $K_3Fe(CN)_6/K_4Fe(CN)_6$	1.43	0.55% (3.95%)	Advantage: relative high efficiency and power density in TGCs Limitation: low efficiency	3
Temperature gradient (flowing electrolytes in symmetric redox reactions at different temperatures)	RFB	Electrode: carbon cloth Flow electrolyte: $[Fe(CN)_6]^{3-}/[Fe(CN)_6]^{4-}$ and V^{3+}/V^{2+}	3.0	1.8% (15%)	Advantage: relative high efficiency, high α , continuous operation across a broad range of temperatures Limitation: relative high cost in electrolyte and operation	4
Temperature cycle (operating between hot and cold reservoirs alternating in a thermal cycle; charging and discharging at different temperatures)	TREC	Electrode: CuHCF and Cu Electrolyte: $NaNO_3$ and $Cu(NO_3)_2$	1.2	3.7% (25%)	Advantages: high efficiency Limitation: the need of external electricity for charging process and the use of ionic selective membrane	5
		Electrode: NiHCF and Ag/AgCl Electrolyte: KCl	0.74	1.6% (13%)	Advantages: simple pouch cell structure without using ionic selective membrane Limitation: the need of external electricity for charging process, relative low efficiency	6
		Electrode: $KFe^{II}Fe^{III}(CN)_6$ and $K_3Fe(CN)_6/K_4Fe(CN)_6$ with carbon cloth Electrolyte: KNO_3	1.45	0.72% (6.0%)	Advantages: charging-free system Limitation: low efficiency, the use of ionic selective membrane	7
	TRAB	Electrode: Cu Electrolyte: $Cu(NO_3)_2/NH_4NO_3$	–	0.86% (6.1%)	Advantages: high power density, low cost Limitation: ammonia stream is a serious concern regarding the leakage, stability, and safety	8
		Electrode: Cu Flow electrolyte: $Cu(NO_3)_2/NH_4NO_3$	–	0.70% (5.0%)	Advantages: high power density, continuous operation Limitation: ammonia stream is a serious concern regarding the leakage, stability, and safety	9
Temperature cycle (thermal-charging and discharging under high temperature and self-regenerated at low temperature)	DTCC	Electrode: GO/PtNPs and PANI Electrolyte: $FeCl_2/FeCl_3$	5.0	5.84% (44.5%)	Advantages: both high efficiency and power density, high α , low cost and easy operation mode, simple pouch cell structure without using ionic selective membrane Limitation: the degradation of long-term cycling	This work

2. In addition, in Figure S2, it shows a shoulder of potential increase to reach the stable platform, which means the active materials (Go and/or PANi) has took time to changes their states due to the potential difference.

Response: We thank the reviewer for this observation. We agree the reviewer that GO and PANI would take time for their corresponding thermal-responses. In this work, we used a sandwiched heating controller where our pouch cell is located between two thermoelectric modules with heat sinks at each side. Thermocouples are placed between the pouch cell and thermoelectric modules. Fig. R3.1 show the temperature profile in our experiment, which takes around 100s to reach a stable temperature. For the cell measurement in Supplementary Fig. 2, it takes a longer time around 150s to reach the stable voltage because of the experimental temperature profile, heat transport and the thermal-response of GO and PANI. We add this temperature profile in Supplementary Fig. 2e.

Fig. R3.1| Temperature profiles in thermal discharging

3. Even though the authors claimed that the device could be regenerated, no real cycling performance were provided with proper materials and electrode characterization before and after cycling.

Response: We thank the reviewer for his/her comment and suggestion. As mentioned in our main text, DTCC worked for 20 cycles as shown in Fig. 3c. To confirm the regeneration of PANI, we did the X-ray photoelectron spectroscopy(XPS) of PANI before, after discharging and after regeneration as we discussed in Supplementary Note 7 and Supplementary Fig. 10. Here we duplicate the Supplementary Fig. 10 below as Fig R3.2 to make it convenient for the reviewer to read. It shows the XPS characterization of the PANI samples for as-prepared, after-discharging and after-regeneration. XPS result of the as-prepared leucoemeraldine PANI shows the presence of nitrogen (N1s, ~400eV), which consists of amine nitrogen (=N-, 398.8 eV), imine nitrogen

(-NH^- , 400 eV) and positively charged nitrogen (-NH^+ , 402 eV). After discharging, the PANI was oxidized and thus -NH^- and -NH^+ were transformed to =N- ; the percentage of -NH^- decreased from 82% to 79% and the percentage of -NH^+ decreased from 9% to 8%, while the percentage of =N- increased from 9% to 13%. To regenerate the electrode, the used PANI (after discharging) was immersed in a 0.5 M $\text{FeCl}_2/0.1$ M HCl solution (similar to the condition of electrolyte after discharging) for 60 minutes. The content of -NH^- increased to 83% and =N- decreased to 7%, evidencing the chemical reduction to recover the PANI. Meanwhile, the color of the solution was changed from green (Fe^{2+}) to yellow (Fe^{3+}) after the reaction.

Fig. R3.2 | XPS of N1s on **a**, as-prepared PANI, **b**, PANI after one cycle of thermal charging and electrical discharging and **c**, Used PANI after immersing in 0.5M $\text{FeCl}_2/0.1$ M HCl.

Other than XPS, as suggested by reviewer for material and electrode characterization, we also add an experiment of Fourier-transform infrared spectroscopy (FTIR) to characterize materials and electrodes before and after cycling. Fig. R3.3 shows the FTIR spectra of GO/PtNPs electrodes and PANI electrodes before cycles, after 10 cycles and after 20 cycles. GO/PtNPs electrode and PANI electrode remained unchanged in the first 10 cycles, but peaks were changed and shifted after 20 cycles. As described in XPS results, PANI is oxidized during electrical discharging and refreshed at room temperature via self-regeneration, which can also be confirmed by the unchanged FTIR features after 10 cycle. The IR bands at 1568 cm^{-1} and 1494 cm^{-1} are assigned to C=C stretching vibrations of benzenoid and aromatic ring while that at 1634 cm^{-1} is attributed to C=N bond and C=C bond vibration. After 20 cycles, the peak shift from 1568 cm^{-1} to 1634 cm^{-1} corresponds to a transformation from amine link to imine link of PANI, indicating the oxidation of PANI. Therefore, the PANI is still gradually oxidized and consumed after a long-term cycling. The peaks of GO become smaller due to the consumption of oxygen functional groups or the materials peeled off during cycles (refer to Supplementary Fig. 14). One possible solution is to take the cell apart after tens of cycles and reassemble it after regenerating electrodes via chemical treatments^{3,4}.

Fig. R3.3 | FTIR spectra of **a**, GO/PtNPs electrodes and **b**, PANI electrodes before cycling, after 10 cycles and after 20 cycles.

The abovementioned results have been newly added as Supplementary Note 8 with Supplementary Fig. 13. The related sentences have been revised correspondingly in the revised manuscript (highlighted in blue in line 212 on page 10).

3. Therefore, I would suggest the authors to conduct a proper major revision of current manuscript before it is considered for publication.

Response: We have thoroughly addressed every comment made by the reviewers and revised the manuscript accordingly. We hope that our response will be satisfactory.

References

1. Izatt, R. M., Watt, G. D., Bartholomew, C. H. & Christensen, J. J. Calorimetric study of Prussian blue and Turnbull's blue formation. *Inorg. Chem.* **9**, 2019-2021 (1970).
2. Butoi, B., Groza, A., Dinca, P., Balan, A. & Barna, V. Morphological and structural analysis of polyaniline and poly (o-anisidine) layers generated in a DC glow discharge plasma by using an oblique angle electrode deposition configuration. *Polymers* **9**, 732 (2017).
3. Zhao, H. *et al.* Oxygen plasma-treated graphene oxide surface functionalization for sensitivity enhancement of thin-film piezoelectric acoustic gas sensors. *ACS Appl. Mater. Interfaces* **9**, 40774-40781 (2017).
4. Moon, D. K., Ezuka, M., Maruyama, T., Osakada, K. & Yamamoto, T. Chemical reduction of the emeraldine base of polyaniline by reducing agents and its kinetic study. *Die Makromolekulare Chemie: Macromolecular Chemistry and Physics* **194**, 3149-3155 (1993).

Reviewers' comments:

Reviewer #1 (Remarks to the Author):

I reviewed the responses from the authors and I am satisfied with these responses.

Reviewer #2 (Remarks to the Author):

Much has been improved through the revision process, but it still has to be revised or unclear. Please refer to the following questions from this reviewer.

1. The DTCC proposed here determines the total voltage of the cell by connecting the pseudocapacitive and the thermogalvanic effects in series. If so, the output current due to both effects will also be connected in series, and the total output current will be limited by the voltage source with low current output. However, the authors seem to explain that the cell voltage in the revised text is generated by connecting two effects in series, and the current is output by connecting two effects in parallel. Provide an equivalent circuit for the DTCC and explain clearly the voltage and current deliveries from the DTCC.

2. The authors detailed in supporting information that the performance degradation of the DTCC due to repeated driving cycles. The DTCC system is a very creative and exciting study, and its academic significance is thought to be very large, but the drawbacks of low repeatability are obvious. The reason for the degradation should be clearly stated in the text.

3. There is still some uncertainty in efficiency calculations. The W_{eff} calculations provided by the authors include the electrical work covered by the initial built-in V_{oc} , which is not actually available in the DTCC operation. The equation of $\frac{1}{2} * C * (V_i - \Delta V_0)^2$ (not $\frac{1}{2} * C * V_i^2 - \frac{1}{2} * C * \Delta V_0^2$) should be used to calculate the W_{eff} , as considering the DTCC cycle. This part is a very important issue directly linked to the efficiency calculations and thus, the authors must clarify it. Please refer to the V-C curve shown attached.

Reviewer #3 (Remarks to the Author):

The authors has addressed reviewers' comments. Therefore, I would suggest it to be considered / accepted for publication at its current stage.

In addition, I would also suggest the authors add one quality perspective paper recently published by JOULE, "Body Heat Powers Future Electronic Skins, Joule 3, 6, 1399-1403."

Revision/Rebuttal Report for NCOMMS-19-10626A

Direct Thermal Charging Cell for Converting Low-grade Heat to Electricity

Xun Wang, Yu-Ting Huang, Chang Liu, Kaiyu Mu, Ka Ho Li, Sijia Wang, Yuan Yang, Lei Wang, Chia-Hung Su, Shien-Ping Feng

We thank the editor for handling the manuscript review and the reviewers for their detailed and insightful comments. The manuscript has been revised to reflect the comments of editor and all reviewers. A detailed revision/rebuttal report is included below, and all revised text in the manuscript and supplementary material is marked in blue for convenience.

Reviewer #1

I reviewed the responses from the authors and I am satisfied with these responses.

Response: We appreciate the reviewer's comment on the publication of this work.

Reviewer #2

Much has been improved through the revision process, but it still has to be revised or unclear. Please refer to the following questions from this reviewer.

1. The DTCC proposed here determines the total voltage of the cell by connecting the pseudocapacitive and the thermogalvanic effects in series. If so, the output current due to both effects will also be connected in series, and the total output current will be limited by the voltage source with low current output. However, the authors seem to explain that the cell voltage in the revised text is generated by connecting two effects in series, and the current is output by connecting two effects in parallel. Provide an equivalent circuit for the DTCC and explain clearly the voltage and current deliveries from the DTCC.

Response: We thank the reviewer for his/her positive comment on the revised manuscript. In DTCC, the voltage and current are from different sources. Here we provide the equivalent circuit to explain the voltage and current delivered from DTCC. The Randles circuit is usually used to express the electrochemical reactions occurred at solid-liquid interface, where the electrons can go through the interface of the working electrode by capacitive current or by Faraday current caused by an electrochemical reaction. Figure R1 shows the equivalent circuit of GO|KCl|PANI and GO/PtNPs|Fe²⁺/Fe³⁺|PANI DTCC at cathode-electrolyte interface. For GO|KCl|PANI cell, the pseudocapacitive behaviour of GO can be expressed by Randles circuit consisting of a parallel combination of pseudocapacitance (C_p) and Faradaic resistance (R_p) connected in series with a charge transfer resistance (R_{CT}) of KCl electrolyte. For GO/PtNPs|Fe²⁺/Fe³⁺|PANI DTCC cell, the Randles circuit of GO is connected to another Randles circuit inclusive of constant phase element (CPE) and R_{CT} of Fe²⁺/Fe³⁺ redox electrolyte. The electrochemical redox reaction, such as Fe^{2+/3+}, rarely shows an ideal impedance response so that the CPE typically reflects a distribution of reactivity that is commonly represented in equivalent electrical circuits.

Open circuit thermal-charging process: When heating in open circuit condition, DTCC generates voltage via temperature-induced pseudocapacitive effect of GO and thermogalvanic effect of $\text{Fe}^{2+}/\text{Fe}^{3+}$, corresponding to C_p and CPE respectively. If we assume that the thermal-induced charge (Q_{charge}) is nearly fixed at the GO-aqueous interface at a certain temperature, the capacitance would become less when C_p and CPE are connected in series, which causes the increase of voltage ($Q_{\text{charge}}=CV$). It is noted that the thermal-induced charge is relatively small as compared with the total charge during discharging process.

Electrical discharging process: When external circuit is connected, the generated voltage can trigger and drive the oxidation of PANI so that the electrons are released from PANI anode to GO/PtNPs cathode. During discharging, the majority of electrons is provided by the oxidation of PANI, which can flow through cathode-electrolyte interface and are then carried by the reduction reaction of Fe^{3+} to Fe^{2+} . As compared with the large R_{CT} in GO|KCl|PANI cell, the redox-active electrolyte of $\text{Fe}^{2+}/\text{Fe}^{3+}$ has a much lower R_{CT} to greatly facilitate the interfacial charge transfer and thus produces a much higher current. Therefore, the increase of total charge ($Q_{\text{discharge}} \gg Q_{\text{charge}}$) significantly enhance the total discharge capacity during discharging process.

The thermal-induced voltage of DTCC is built by the thermo-pseudocapacitive and the thermogalvanic effects connecting in series in open circuit condition, while the current of DTCC (driven by the voltage) are mainly sourced from the oxidation of PANI anode with the aid of the reduction reaction of Fe^{3+} to Fe^{2+} in electrolyte when the circuit is connected. Therefore, the total output current is not limited by the voltage source in cathode side because a large amount of electrons is provided by continuously oxidizing PANI anode after connecting the circuit. The description of equivalent circuit has been added in Supplementary Note 5 and Supplementary Fig. 8.

Fig. R1| Equivalent circuit for **a**, GO|KCl|PANI and **b**, GO/PtNPs| $\text{Fe}^{2+}/\text{Fe}^{3+}$ |PANI DTCC at cathode-electrolyte interface in thermal charging and electrical discharging process, where C_p is pseudocapacitance, R_p is a related faradic resistance of pseudocapacitive reaction, R_{CT} is charge transfer resistance between cathode and electrolyte and CPE stands for the constant phase element for $\text{Fe}^{2+}/\text{Fe}^{3+}$ redox reaction.

2. The authors detailed in supporting information that the performance degradation of the DTCC due to repeated driving cycles. The DTCC system is a very creative and exciting study, and its academic significance is thought to be very large, but the drawbacks of low repeatability are obvious. The reason for the degradation should be clearly stated in the text.

Response: We thank the reviewer for his/her positive comment and suggestion. As we mentioned in Supplementary Note 8, it is noted that the oxygen functional groups of GO are still consumed and PANI is gradually oxidized at each cycle until their depletion, which would limit the long-term cycle number (Supplementary Note 8, Supplementary Fig. 13 and 14). Therefore, further research with the optimization of electrode materials, electrolyte, and cell packaging is needed to improve the cyclability of DTCCs. The information has been added in the main text in Page 10 Line 213. Now, we are working on the modification of electrode materials to achieve a stable and long cycling performance. Hopefully, the result can be published in the near future.

3. There is still some uncertainty in efficiency calculations. The W_{eff} calculations provided by the authors include the electrical work covered by the initial built-in V_{oc} , which is not actually available in the DTCC operation. The equation of $1/2 * C * (V_i - \Delta V_0)^2$ (not $1/2 * C * V_i^2 - 1/2 * C * \Delta V_0^2$) should be used to calculate the W_{eff} , as considering the DTCC cycle. This part is a very important issue directly linked to the efficiency calculations and thus, the authors must clarify it. Please refer to the V-C curve shown attached.

Response: We thank the reviewer for his/her comments. Here, we would like to compare DTCC with hybrid supercapacitor. The discharging process of DTCC stopped at the initial built-in ΔV_0 , which is quite similar to the discharging process of hybrid supercapacitor using asymmetric electrodes. In the published papers for hybrid supercapacitor (*Chem. Soc. Rev.* **45**, 5925, 2016; *Langmuir* **33**, 9407, 2017; *Nat. Mater.* **17**, 167, 2018; *Chem. Rev. (Washington, DC, U. S.)* **118**, 6457, 2018), the built-in voltage actually boosts up the total operating voltage so that the total work output can be improved by using asymmetric electrodes¹⁻⁴. Here we copy the Fig. 15 in ref. 1 (*Chem. Soc. Rev.* **45**, 5925, 2016) as Fig. R2 below to make it convenient for the reviewer to read. As seen, the electrical work covered by the initial built-in voltage is included in the hybrid supercapacitor system (Fig. R2b).

Fig. R2 | Schematic of discharge profile of (a) EDLC and (b) hybrid supercapacitor (reproduce from ref. ¹).

The abovementioned calculation of effective output energy (W_{eff}) is the same as the equation we used in the article

$$W_{eff} = \int_0^{\Delta Q} V_{dis} dQ \approx \frac{1}{2} CV_i^2 - \frac{1}{2} C(\Delta V_0)^2$$

where V_i is used instead of V_{oc} due to the IR drop, Q is the discharging capacity, C is the capacitance of DTCC. Similar to the hybrid supercapacitor, the total voltage of DTCC is actually boosted up by the built-in ΔV_0 ($V_i > \Delta V_0 > 0$) so that the all covered black shade area based on the integration of voltage and capacity should be the real output work of DTCC (Fig. R3). The device with the capacitance of C at the voltage of V_i stores the total energy of $1/2 CV_i^2$; after discharging to ΔV_0 , the remaining energy in the device is $1/2 C(\Delta V_0)^2$. Hence, the output work is $1/2 CV_i^2 - 1/2 C(\Delta V_0)^2$ (total energy – remaining energy). Our calculation method is consistent with the published papers in hybrid supercapacitor. Actually, one purpose of using asymmetric electrodes in hybrid system is to boost up the output voltage by the built-in voltage to increase the output work.

Fig. R3 | Full cell voltage v.s. capacity in discharging process

In addition, to prove and confirm the real energy output of DTCC, we used one DTCC to charge a 4.7F supercapacitor (Goldcap, Panasonic) and checked how much energy can be released from DTCC and then stored in the supercapacitor. DTCC was heated up to 70°C and then connected to the supercapacitor. During the process, the voltage of DTCC decreased and the voltage of supercapacitor increased until both voltages reached the same value of around 0.245V (Fig. R4). In Fig. R4a, the energy charged to the supercapacitor can be simply calculated by $W_{SC} = 1/2CV^2 = 1/2(4.7)(0.245)^2 = 0.141$ J. On the other hand, Fig. R4b is the constant-current discharging curve of DTCC with the end voltage V_e of 0.245V. The energy output (W_{eff}) calculated by $1/2CV_i^2 - 1/2C(V_e)^2$ is 0.168 J, which is equal to W_1 (green area $>\Delta V_0$) + W_2 (orange area $<\Delta V_0$) = 0.109 J + 0.059 = 0.168 J.

As seen, the green area above ΔV_0 (W_1) is only 0.109 J (exclude W_2 covered by ΔV_0), which is much lower than the real energy of 0.141 J stored in supercapacitor. Hence, this experiment demonstrates that the built-in area actually contributes to the effective energy output of DTCC, which should be included in the overall device performance. As noted, the stored energy of 0.141 J in supercapacitor is lower than the total output energy of 0.168 J in DTCC, which usually comes from the energy loss by conduct wires, contact resistance and self-discharging of supercapacitor.

In other word, the effective energy output would be changed if we employ two electrodes with different built-in voltages. Now, we are tuning the working function of GO and PANI and found that the built-in voltage indeed influences the overall voltage output and device performance. We hope the systematic study can be published in the future. If we calculate by using $1/2C(V-\Delta V_0)^2$, all the device performances (no matter how large or small the built-in voltage is) will be the same as long as the device has a similar capacitance and thermal-voltage rise, which can not reflect the real and effective energy output of DTCC or other hybrid systems. We hope the answer will be satisfactory for the reason we include the built-in voltage in the efficiency calculation.

Fig. R4 | **a**, Voltage profile of 4.7F supercapacitor charged by DTCC operating at 70°C; **b**, DTCC cell voltage versus capacity when discharged at a constant current of 0.1 mA under 70°C.

Reviewer #3

The authors has addressed reviewers' comments. Therefore, I would suggest it to be considered / accepted for publication at its current stage.

In addition, I would also suggest the authors add one quality perspective paper recently published by JOULE, "Body Heat Powers Future Electronic Skins, Joule 3, 6, 1399-1403."

Response: We appreciate the reviewer's comment on the publication of this work. The paper, "Body Heat Powers Future Electronic Skins, Joule 3, 6, 1399-1403.", presents the recent advances and challenges in E-skins, and the conceptual design of E-skin and thermoelectric device integration to harvest body heat, which is an inspiring work for us to possibly employ the DTCC to harvest body heat. This paper has been cited as reference 42.

References

1. Wang, Y., Song, Y. & Xia, Y. Electrochemical capacitors: mechanism, materials, systems, characterization and applications. *Chem. Soc. Rev.* **45**, 5925-5950 (2016).
2. Jeżowski, P. *et al.* Safe and recyclable lithium-ion capacitors using sacrificial organic lithium salt. *Nat. Mater.* **17**, 167 (2018).
3. Lai, C.-H. *et al.* Designing pseudocapacitance for Nb₂O₅/carbide-derived carbon electrodes and hybrid devices. *Langmuir* **33**, 9407-9415 (2017).
4. Ding, J., Hu, W., Paek, E. & Mitlin, D. Review of hybrid ion capacitors: from aqueous to lithium to sodium. *Chem. Rev. (Washington, DC, U. S.)* **118**, 6457-6498 (2018).

Reviewers' comments:

Reviewer #2 (Remarks to the Author):

This reviewer agrees with the authors' answers to questions 1 and 2. However, I still can not agree with the authors' answer to question 3 regarding efficiency calculation.

This work is basically a new study of a device that converts thermal energy into electrical energy, whose efficiency is defined as the output electrical energy for the input thermal energy. Considering the thermal cycle of the DTCC, this reviewer does believe that the output electrical energy should be calculated as $1/2 * C * (V_i - \Delta V_0)^2$.

As clearly shown in the driving mechanism of Fig. 1 and as evidenced in Figure 16 (b), the cell voltage initiated at the built-in voltage (ΔV_0) returns to the initial ΔV_0 after the thermal cycle. Therefore, the area under ΔV_0 in the CV curve should be excluded from the cycle efficiency calculation. If the authors would estimate the electrical energy of the entire region as an output power (as shown in Fig. R3), the DTCC cell should take into account the "input energy" for the efficiency calculation to return to the ΔV_0 from zero volt. Otherwise, this DTCC cycle violates energy conservation laws and is thermodynamically inoperable.

Despite these errors in efficiency calculations (at least in this review, I believe so), it is undeniable that this research has a very novel and unique approach to the development of thermal energy conversion devices. Also, from the measurement results presented in the paper, it is judged that the output performance is quite attractive when the reviewer roughly calculates the efficiency. Therefore, if the authors could report a collected performance metric through a clear calculation of efficiency, the reviewer would recommend it to be acceptable in this journal, just like other reviewers. However, if authors do not agree with the claims of this reviewer, I can no longer review it. Please acknowledge this point.

Revision/Rebuttal Report for NCOMMS-19-10626B

Direct Thermal Charging Cell for Converting Low-grade Heat to Electricity

Xun Wang, Yu-Ting Huang, Chang Liu, Kaiyu Mu, Ka Ho Li, Sijia Wang, Yuan Yang, Lei Wang, Chia-Hung Su, Shien-Ping Feng

We thank the editor for handling the manuscript review and the reviewer for his/her detailed and insightful comments. The manuscript has been revised to reflect the comments of editor and reviewer. A detailed revision/rebuttal report is included below, and all revised text in the manuscript and supplementary material is marked in blue for convenience.

Reviewer #2

*This reviewer agrees with the authors' answers to questions 1 and 2. However, I still can not agree with the authors' answer to question 3 regarding efficiency calculation. This work is basically a new study of a device that converts thermal energy into electrical energy, whose efficiency is defined as the output electrical energy for the input thermal energy. Considering the thermal cycle of the DTCC, this reviewer does believe that the output electrical energy should be calculated as $1/2 * C * (V_i - \Delta V_0)^2$. As clearly shown in the driving mechanism of Fig. 1 and as evidenced in Figure 16 (b), the cell voltage initiated at the built-in voltage (ΔV_0) returns to the initial ΔV_0 after the thermal cycle. Therefore, the area under ΔV_0 in the CV curve should be excluded from the cycle efficiency calculation. If the authors would estimate the electrical energy of the entire region as an output power (as shown in Fig. R3), the DTCC cell should take into account the "input energy" for the efficiency calculation to return to the ΔV_0 from zero volt. Otherwise, this DTCC cycle violates energy conservation laws and is thermodynamically inoperable. Despite these errors in efficiency calculations (at least in this review, I believe so), it is undeniable that this research has a very novel and unique approach to the development of thermal energy conversion devices. Also, from the measurement results presented in the paper, it is judged that the output performance is quite attractive when the reviewer roughly calculates the efficiency. Therefore, if the authors could report a collected performance metric through a clear calculation of efficiency, the reviewer would recommend it to be acceptable in this journal, just like other reviewers. However, if authors do not agree with the claims of this reviewer, I can no longer review it. Please acknowledge this point.*

Response: We thank the reviewer for his/her comment. After careful consideration, we agree the reviewer that the output electric energy should be calculated by $W = \frac{1}{2} C (V_i - \Delta V_0)^2$ to exclude the energy area under ΔV_0 which is related to the intrinsic chemical potential imbalance of electrodes and does not contribute to thermo-electrochemical effect. After this correction, the efficiency of DTCC reaches 2.8% at 70°C (21.4% of η_{Carnot}) and 3.52% at 90° ($\eta_{\text{Carnot}}=17.9\%$) which is lower than the previous calculation, but the efficiency and power density are still at the forefront performance compared with the existing TECs and TEs in the low-grade heat regime. Here we duplicate the revised Fig. 4a and 4b as Fig.R1 to make it convenient for the reviewer to

read. The manuscript, supplementary note 10 and supplementary table 2 are revised accordingly and highlighted in blue.

Fig. R1| Device performance and demonstration. **a**, Heat-to-electricity conversion efficiency (η_E) versus temperature differential, and **b**, ratio of η_E to η_{Carnot} versus volumetric power density for DTCCs and the best-reported TECs including TGCs (blue triangle), RFB-based TEC (violet diamond), TRABs (green square), and TREC (orange circle).

REVIEWERS' COMMENTS:

Reviewer #2 (Remarks to the Author):

The manuscript demonstrates an interesting study on harvesting waste thermal energy. The characterizations are now clear and the claimed device is promising, therefore, this work is acceptable for publishing in Nature communications.

Revision/Rebuttal Report for NCOMMS-19-10626C

Direct Thermal Charging Cell for Converting Low-grade Heat to Electricity

Xun Wang, Yu-Ting Huang, Chang Liu, Kaiyu Mu, Ka Ho Li, Sijia Wang, Yuan Yang, Lei Wang, Chia-Hung Su, Shien-Ping Feng

We thank the editor and reviewers for the acceptance on this work to be published.

Reviewer #2

The manuscript demonstrates an interesting study on harvesting waste thermal energy. The characterizations are now clear and the claimed device is promising, therefore, this work is acceptable for publishing in Nature communications.

Response: We appreciate the reviewer for the recommendation for publishing this work.